# NorMuon: Making Muon More Efficient and Scalable

**Zichong Li** [* 1]  **Liming Liu** [* 1]  **Chen Liang** [2]  **Weizhu Chen** [2]  **Tuo Zhao** [1]

## Abstract

The choice of optimizer significantly impacts the training efficiency and computational costs of large language models (LLMs). Recently, the Muon optimizer has demonstrated promising results by orthogonalizing parameter updates, improving optimization geometry through better conditioning. Despite Muon's emergence as a candidate successor to Adam, the potential for jointly leveraging their strengths—has not been systematically explored. In this work, we bridge this gap by proposing NorMuon (Neuron-wise Normalized Muon), an optimizer that synergistically combines orthogonalization with neuron-level adaptive learning rates. Our analysis reveals that while Muon effectively reduces condition numbers, the resulting updates exhibit highly non-uniform neuron norms, causing certain neurons to dominate the optimization process. Nor-Muon addresses this imbalance by maintaining second-moment statistics for each neuron and applying row-wise normalization after orthogonalization, ensuring balanced parameter utilization while preserving Muon's conditioning benefits. To enable practical deployment at scale, we develop an efficient distributed implementation under the FSDP2 framework that distributes orthogonalization computations across devices. Experiments across multiple model scales demonstrate that NorMuon consistently outperforms both AdamW and Muon, achieving a 21.74% reduction in training steps relative to AdamW and an 11.31 percentage-point larger efficiency gain than Muon on 1.1B pretraining. Results suggest that orthogonalization and adaptive learning rates are complementary rather than competing, opening new avenues for optimizer design in large-scale deep learning.

---

[*]Equal contribution [1]Georgia Institute of Technology [2]Microsoft. Correspondence to: Zichong Li <zli911@gatech.edu>, Liming Liu <lliu606@gatech.edu>.

*Proceedings of the 43rd International Conference on Machine Learning*, Seoul, South Korea. PMLR 306, 2026. Copyright 2026 by the author(s).

## 1 Introduction

Training efficiency remains a central challenge in scaling large language models (LLMs) (Sachdeva et al., 2024; Wan et al., 2023), where optimizer choice directly impacts convergence speed, computational requirements, and ultimately, the feasibility of training at scale (Jordan et al., 2024b; Wen et al., 2025). The community standard, Adam (Kingma and Ba, 2015), achieves robust performance through coordinate-wise preconditioning: dynamically adjusting learning rates for each parameter based on the second moment of its gradient history. While this per-coordinate adaptivity is computationally efficient and generally stable, it suffers from a fundamental limitation—it treats each parameter independently, ignoring the rich geometric structure and cross-coordinate dependencies inherent in neural network layers.

Recent advances have sought to address this limitation through various approaches to capturing cross-coordinate structure. Adam-mini (Zhang et al., 2025) exploits the near-block-diagonal Hessian structure of neural networks by applying adaptive learning rates to parameter blocks (e.g. each neuron) rather than individual coordinates. More ambitious second-order methods, such as Shampoo (Gupta et al., 2018) and SOAP (Vyas et al., 2024), employ full-matrix preconditioning through singular value decomposition to capture curvature information and parameter inter-dependencies. However, these approaches incur substantial memory and communication overhead, while introducing hyperparameter sensitivity that limits their practical adoption at scale.

Recently, Muon (Jordan et al., 2024b) has emerged as a compelling middle ground, applying truncated Newton-Schulz iterations to approximate the orthogonal polar factor of momentum matrices. This approach yields matrix-wise orthogonalized updates that improve conditioning while maintaining modest computational overhead and approximately half the memory consumption of Adam, demonstrating promising results in LLM training.

These optimizers fundamentally differ in their preconditioning granularity and objectives. Adam and Adam-mini, when considered without exponential moving averages (EMAs), apply $L^2$ normalization at the per-coordinate and per-neuron levels respectively, adjusting learning rates

while preserving update signs. In contrast, the idealized version of Shampoo and Muon operate at the per-matrix level, actively orthogonalizing parameter updates.

The varied preconditioning strategies employed by these optimizers raise an important question: *Are different forms of preconditioning inherently conflicting, or can they be combined in a way that yields complementary benefits?*

To investigate this, we analyzed key properties of the update matrices from different optimizers during pretraining of a 1.1B-parameter Transformer model, examining both singular value distributions and per-neuron norms. As illustrated in Figure 1a, raw momentum accumulates updates with extremely high condition numbers, indicating that certain directions dominate while leaving other parameters underutilized. AdamW produces moderately more balanced singular values, though the improvement remains limited. In contrast, Muon's approximate orthogonalization successfully addresses this conditioning issue, yielding well-balanced singular values across the spectrum. However, examining per-neuron update norms (Figure 1b) reveals a complementary perspective. AdamW demonstrates superior performance in reducing variance across per-neuron update norms compared to SGD momentum. Conversely, while Muon's orthogonalization effectively improves matrix-level conditioning, the per-neuron update norms exhibit high variance, with some neurons receiving disproportionately large updates relative to others.

This observation motivates our key insight: while Muon's orthogonalization effectively reduces the condition number of updates, the remaining high variance in neuron norms still creates an imbalanced learning dynamic, potentially leading to inefficient parameter usage. Drawing inspiration from Adam-mini's success (Zhang et al., 2025) with per-neuron adaptive learning rates, we propose to incorporate second moment estimate to normalize these disparate scales and ensure more balanced parameter updates. Our method, **NorMuon**, augments Muon's orthogonalization with neuron-wise adaptive learning rates computed from accumulated second-moment statistics. As demonstrated in our analysis, NorMuon yields updates with both low condition numbers (Figure 1a) and uniform neuron norms (Figure 1b), thereby combining the advantages of Muon and AdamW and achieving more balanced utilization of the network's representational capacity.

Beyond algorithmic innovation, the distributed implementation of orthogonalization-based optimizers remains relatively underexplored in the literature. To enable training at larger scales, we develop a distributed version of NorMuon compatible with the FSDP2 framework (Feng et al., 2022). While previous work on distributed Muon (Liu et al., 2025a) was implemented using ZeRO-1 with Megatron-LM (Rajbhandari et al., 2020; Shoeybi et al.,

2019), FSDP2 can offer greater flexibility and memory efficiency. However, direct adaptation of the previous distributed approach to FSDP2 would result in extensive replicated computation, as FSDP2 shards nearly all parameters across devices. Our implementation addresses this challenge by distributing orthogonalization computation across devices, eliminating redundant calculations while maintaining load balance. Furthermore, we leverage FSDP2's row-wise parameter sharding to enable efficient neuron-wise normalization without incurring additional communication overhead.

In summary, our contributions are threefold:

• We propose NorMuon, a simple and effective optimizer that combines Muon's orthogonalization with neuron-wise adaptive learning rates. NorMuon maintains uniform neuron norms to ensure balanced parameter utilization while preserving the low condition number achieved by Muon's orthogonalization.

• We develop an efficient distributed implementation under FSDP2 framework. By carefully orchestrating sharded optimizer states, we gather updated momentum and distribute Muon orthogonalization computation uniformly across GPUs, achieving optimal memory efficiency with manageable communication and computational overhead.

• Through extensive experiments across multiple scales of LLM pretraining, we show that orthogonalization and blockwise adaptive learning rates are complementary rather than conflicting, with their combination yielding superior training dynamics compared to either approach in isolation.

## 2 Related Works and Background

### 2.1 Related Works

**Adaptive Gradient Methods**. The introduction of per-parameter adaptive learning rates has been instrumental in training deep networks. Optimizers such as AdaGrad (Duchi et al., 2011), RMSProp (Hinton, 2012), Adam (Kingma and Ba, 2015) and AdamW (Loshchilov and Hutter, 2017) use first- and second-moment estimates to adjust each weight's step size individually. This coordinate-wise preconditioning improves stability and convergence in heterogeneous settings, and has become the de facto standard for LLM training. However, treating each weight independently ignores the underlying structure of neural network layers and incurs high memory overhead by storing two extra tensors per parameter. This memory cost motivated techniques like AdaFactor (Shazeer and Stern, 2018), which factorizes the second-moment accumulator across rows and columns to reduce memory. Similarly, Adam-mini (Zhang et al., 2025) partitions parameters into blocks (e.g. each neuron's weights) and assigns a single learn-

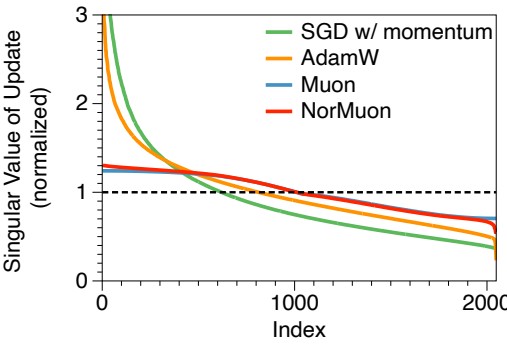

*(a)* Singular value distribution of update directions

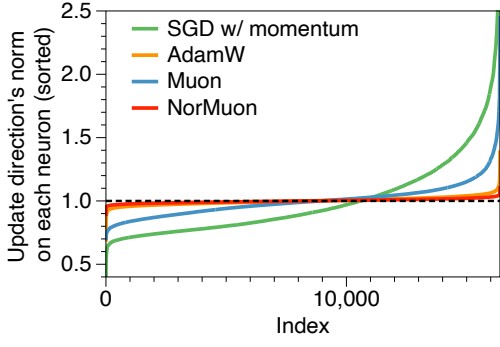

*(b)* Per-neuron update norm

*Figure 1.* Analysis of optimization geometry during 1.1B model pretraining. We examine the up-projection matrix in the 8th layer's MLP at the middle checkpoint. (a) Singular value distribution reveals that raw momentum and AdamW's update exhibit high condition numbers. Muon's orthogonalization effectively eliminates this imbalance. (b) Despite Muon's improved conditioning, the $L^2$ norm of individual neuron updates still shows high variance. AdamW achieves much more uniform per-neuron norms. Our proposed NorMuon maintains Muon's low condition number while normalizing neuron contributions.

ing rate to each block, matching AdamW's performance on different model sizes, while halving memory cost. Ga-Lore (Zhao et al., 2024) maintains momentum in a low-rank subspace derived from the SVD of gradients, although its effectiveness diminishes for long sequence lengths (Liu et al., 2025b). Lion (Chen et al., 2023) applies a coordinate-wise signed update, abandoning second moment estimates to achieve memory savings.

**Second-order Methods**. In parallel, other optimizers capture the rich geometry of the loss surface by coupling updates across parameters. K-FAC (Martens and Grosse, 2015) and its variants (Martens et al., 2018; Gao et al., 2021) approximate curvature information beyond individual coordinates, capturing correlations across parameters. Shampoo (Gupta et al., 2018) and its distributed variant (Shi et al., 2023) employ Kronecker-factored preconditioners and have demonstrated strong performance in practice (Dahl et al., 2023). More recently, SOAP (Vyas et al., 2024) establishes a connection between Shampoo and Adafactor and further improves convergence performance. Despite these advances, Shampoo and SOAP incur substantial memory cost and computational overhead, which hinders their applicability at LLM scale.

**Orthogonal and geometry-aware update methods**. Muon (Momentum Orthogonalized by Newton-Schulz, (Jordan et al., 2024b)) represents a recent breakthrough that leverages matrix geometry without the full cost of second-order methods. Muon performs an approximate polar decomposition via Newton-Schulz iterations on the momentum to extract its orthogonal component, while avoiding the storage of full second-moment tensors. Muon thus improves convergence and memory efficiency compared with AdamW in LLM pretraining, demonstrating strong potential for scalable model training (Liu et al., 2025a; Shah

et al., 2025). Earlier and concurrent geometry-aware optimization methods also study spectral or norm-constrained update directions, including preconditioned spectral descent (Carlson et al., 2015) and training with norm-constrained linear minimization oracles (Pethick et al., 2025). These works provide complementary perspectives on how the geometry of matrix-valued layers can shape effective update rules. More recently, Dion (Ahn et al., 2025) extends the orthogonal update paradigm to be more communication- and compute-efficient in distributed settings by using low-rank orthonormalization via amortized power iteration and decoupled momentum buffers.

### 2.2 Background: Muon optimizer

Muon (Jordan et al., 2024b) is an optimizer designed for the 2D weight matrices in neural network hidden layers. The key innovation lies in orthogonalizing the momentum before applying parameter updates, thereby improving the conditioning of the optimization trajectory. Formally, at iteration $t$, given weight matrix $W_{t-1}$, learning rate $\eta_t$, and loss function $L$, Muon maintains a first moment estimate $M_t$ and computes updates as:

$$M_t = \mu \, M_{t-1} + \nabla L(W_{t-1}),$$
$$O_t = \mathrm{NS5}(M_t),$$
$$W_t = W_{t-1} - \eta_t \, O_t,$$

where $M_0 = \mathbf{0}$ and $\mu$ is the momentum coefficient. The critical component is the orthogonalization operator $\mathrm{NS5}(\cdot)$, which aims to approximate the orthogonal projection of the momentum matrix:

$$\mathrm{Ortho}(M) = \arg\min_{O}\{\|O - M\|_F : O^\top O = I \text{ or } OO^\top = I\}.$$

Muon approximates this orthogonalization through a fixed number of Newton-Schulz iterations. Starting with the

Frobenius-normalized momentum $X_0 = M_t/\|M_t\|_F$, the algorithm performs $N$ iterations (typically $N = 5$):

$$X_k = a\, X_{k-1} + b\, (X_{k-1}X_{k-1}^\top)\, X_{k-1} + c\, (X_{k-1}X_{k-1}^\top)^2\, X_{k-1}$$

for $k = 1, \ldots, N$, with the final orthogonalized update $O_t = X_N$. The coefficients $(a, b, c)$ are carefully chosen such that singular values of the update matrix converge toward unity. In practice, Muon is typically applied only to 2D weight matrices in hidden layers, while scalar parameters, bias vectors, embeddings, and unembedding layers continue to use standard optimizers such as Adam.

## 3    Method

In this section, we introduce NorMuon, where we aim to combine Muon's orthogonalization with block-wise adaptive learning rates based on the observation that the approximated orthogonalized updates can experience a high variance on update directions norm of each neuron.

---

**Algorithm 1** NorMuon

---

1: **Input:** Initial weights $\mathbf{W}_0 \in \mathbb{R}^{m \times n}$, loss $L$, learning rate $\eta$, momentum parameters $(\beta_1, \beta_2)$, perturbation parameter $\varepsilon$, weight decay $\lambda$.
2: Initialize $\mathbf{M}_0 \in \mathbb{R}^{m \times n} \leftarrow \mathbf{0}$, $\mathbf{v}_0 \in \mathbb{R}^m \leftarrow \mathbf{0}$
3: **for** $t = 1, 2, \ldots$ **do**
4:   $\mathbf{G}_t \leftarrow \nabla_{\mathbf{W}} L(\mathbf{W}_t)$
5:   $\mathbf{M}_t \leftarrow \beta_1 \mathbf{M}_{t-1} + (1 - \beta_1)\mathbf{G}_t$
6:   $\mathbf{O}_t \leftarrow \mathrm{NS5}(\mathbf{M}_t)$
7:   $\mathbf{v}_t \leftarrow \beta_2 \mathbf{v}_{t-1} + (1 - \beta_2)\, \mathrm{mean}_{\mathrm{cols}}(\mathbf{O}_t \odot \mathbf{O}_t)$
8:   $\mathbf{V}_t \leftarrow \mathrm{ExpandRows}\,(\mathbf{v}_t)$        $(\mathbf{V}_t \in \mathbb{R}^{m \times n})$
9:   $\widehat{\mathbf{O}}_t \leftarrow \mathbf{O}_t \oslash \left(\sqrt{\mathbf{V}_t} + \varepsilon\right)$
10:   $\hat{\eta} = 0.2\eta\sqrt{mn}/\|\widehat{\mathbf{O}}_t\|_F$
11:   $\mathbf{W}_{t+1} \leftarrow \mathbf{W}_t - \eta\lambda\mathbf{W}_t - \hat{\eta}\widehat{\mathbf{O}}_t$
12: **end for**

---

### 3.1    NorMuon

We present our update rule in Algorithm 1. The algorithm maintains two momentum states: the standard first moment estimate $\mathbf{M}_t \in \mathbb{R}^{m \times n}$ used by Muon (line 5), and an averaged second moment estimate $\mathbf{v}_t \in \mathbb{R}^m$ that tracks the squared magnitude of each neuron's update direction (lines 7). Importantly, $\mathbf{v}_t$ requires minimal additional memory overhead, storing only $m$ scalars compared to the $m \times n$ first moment estimate.

At each iteration and given the gradient, we first follow the Muon's update rule that update the first moment estimate and apply Newton-Schulz iteration for orthogonalization (line 4-6), producing $\mathbf{O}_t$ with improved conditioning. Rather than directly using this orthogonalized update, we compute row-wise statistics to capture the per-neuron

update magnitudes. Specifically, we calculate the mean squared value across columns for each row of $\mathbf{O}_t$ (line 7). This statistic is accumulated into our averaged second moment estimate $\mathbf{v}_t$ using exponential moving average with decay rate $\beta_2$. We then apply $\mathbf{v}_t$ for row-wise normalization (line 9). This second moment estimate is similar to Adam-mini's (Zhang et al., 2025) block-wise reduced-dimensional statistics, where we treat each neuron (i.e. each row) as a block. As illustrated in Figure 1, this normalization reduces the variance in update magnitudes across neurons while preserving the favorable conditioning properties.

We observe that after the row-wise normalization the resulting update matrix has a much larger norm. Hence, during the update, we add a learning rate scaling following Jordan et al. (2024b) to keep a similar RMS norm to match Adam's RMS norm (line 10). The constant 0.2 is adopted from prior work as an empirically tuned stability factor; it effectively controls the post-normalization step size and can be further adjusted, which is also tuned in our experiments. Moreover, because this scaling already regulates the RMS norm, we omit the bias-correction commonly used in Adam-style methods, as the correction would be neutralized by the scaling.

We would like to note that in the idealized case where the $\mathbf{O}_t$ is strictly orthogonalized (i.e., not approximated by NS5), the per-neuron norm would be strictly 1 for full-rank matrix with $m \le n$. On these matrices, the neuron-wise normalization would not be beneficial. However, since orthogonalization is approximated in practice, we observe that this normalization remains necessary and helpful even for $m \le n$ matrix (validated in Section 4.1.4).

**Remark on normalization placement.**    Algorithm 1 applies neuron-wise normalization after the Newton-Schulz orthogonalization. For completeness, we also consider an alternative ordering in the ablation study, denoted as *NorMuon (Front)*, where the normalization is applied before orthogonalization. This variant follows the Adam-mini-style convention for its statistic: the row-wise second moment is estimated from the raw gradient,

$$\mathbf{v}_t^{\mathrm{front}} = \beta_2 \mathbf{v}_{t-1}^{\mathrm{front}} + (1 - \beta_2)\, \mathrm{mean}_{\mathrm{cols}}(\mathbf{G}_t \odot \mathbf{G}_t).$$

The statistic is then used to normalize the momentum before NS5,

$$\widetilde{\mathbf{M}}_t = \mathbf{M}_t \oslash \left(\sqrt{\mathrm{ExpandRows}(\mathbf{v}_t^{\mathrm{front}})} + \varepsilon\right),$$

$$\mathbf{U}_t^{\mathrm{front}} = \mathrm{NS5}(\widetilde{\mathbf{M}}_t).$$

We include this variant mainly to examine whether the placement of neuron-wise normalization relative to orthogonalization affects performance. The comparison is reported in Section 4.1.4.

## 3.2 Distributed NorMuon

As LLM training scales larger, distributed training becomes essential for both memory constraints and computational efficiency. We develop a distributed version of NorMuon compatible with the FSDP2 framework (Feng et al., 2022), which employs ZeRO-3 style (Rajbhandari et al., 2020) sharding to partition optimizer states, parameters, and gradients across multiple devices.

While coordinate-wise optimizers like Adam naturally extend to distributed settings, Muon and NorMuon present unique challenges due to Muon's orthogonalization step, which requires access to complete momentum matrices. An existing distributed implementation of Muon (Liu et al., 2025a) gathers the full momentum on all devices and replicates the orthogonalization computation. We avoid such replicated costs by near-uniformly assigning parameters to different devices.

Algorithm 2 presents our distributed implementation. The key modifications from Algorithm 1 are:

• **Efficient Orthogonalization Distribution** (line 5-9): Rather than having all devices compute orthogonalization for all parameters, we first sort the parameter list by matrix size (line 2) to ensure uniform work assignment, where Numel(·) counts the number of elements in each matrix. We then assign each parameter tensor to a specific device using a round-robin scheme. Only the assigned device gathers the full momentum matrix via gather communication and performs the Newton-Schulz orthogonalization (lines 5-8), before scattering the result back to all devices (line 9). This approach eliminates redundant computation while maintaining load balance across devices.

• **Shard-Local Row Normalization** (lines 10-12): An advantage of our design is that the row-wise normalization operates entirely on local shards without additional communication. This is possible because FSDP2 employs row-wise sharding, ensuring that each device holds complete rows of the weight matrix. The computation of row statistics and normalization thus proceed independently.

Importantly, while row-wise sharding aligns naturally with NorMuon, the algorithm is not restricted to this setting. NorMuon can also operate under more complex sharding schemes—e.g. tensor parallelism or column-wise sharding—by aggregating per-row statistics through a lightweight all-reduce before applying the same normalization. The communication overhead relative to Muon remains negligible, since it involves only per-row scalar reductions (<1% of the parameter size). Designing optimal distributed variants of Muon itself under these more intricate sharding configurations is an interesting avenue for future work, and we believe our formulation provides a practical building block for scaling orthogonalization-based optimizers to larger clusters.

---

**Algorithm 2** Distributed NorMuon: one iteration

1: **Input:** Sharded 2D weights $\{\mathbf{W}_{\text{shard}}^{(i)}\}_{i=0,\ldots,N}$, Sharded gradient $\{\mathbf{G}_{\text{shard}}^{(i)}\}_{i=0,\ldots,N}$, learning rate $\eta$, momentum parameters $(\beta_1, \beta_2)$, perturbation parameter $\varepsilon$, weight decay $\lambda$. We omit the initialization of optimizer states for simplicity.

2: $\{\mathbf{W}_{\text{shard}}^{(i)}\}_{i=0,\ldots,N} \leftarrow \text{Sort}(\{\mathbf{W}_{\text{shard}}^{(i)}\}_{i=0,\ldots,N}, \text{key} = \text{Numel}(\cdot))$

3: **for** $i = 0, 1, \ldots, N$ **do**

4: $\quad \mathbf{M}_{\text{shard}}^{(i)} \leftarrow \beta_1 \mathbf{M}_{\text{shard}}^{(i)} + (1 - \beta_1)\mathbf{G}_{\text{shard}}^{(i)}$

5: $\quad$ **if** $i \bmod \text{world size} == \text{current rank}$ **then**

6: $\quad\quad \mathbf{M}^{(i)} \leftarrow \text{Gather}(\mathbf{M}_{\text{shard}}^{(i)})$

7: $\quad\quad \mathbf{O}^{(i)} \leftarrow \text{NS5}(\mathbf{M}^{(i)})$

8: $\quad$ **end if**

9: $\quad \mathbf{O}_{\text{shard}}^{(i)} \leftarrow \text{Scatter}(\mathbf{O}^{(i)})$

10: $\quad \mathbf{v}_{\text{shard}}^{(i)} \leftarrow \beta_2 \mathbf{v}_{\text{shard}}^{(i)} + (1 - \beta_2)\,\text{mean}_{\text{cols}}(\mathbf{O}_{\text{shard}}^{(i)} \odot \mathbf{O}_{\text{shard}}^{(i)})$

11: $\quad \mathbf{V}_{\text{shard}}^{(i)} \leftarrow \text{ExpandRows}\left(\mathbf{v}_{\text{shard}}^{(i)}\right)$

12: $\quad \widehat{\mathbf{O}}_{\text{shard}}^{(i)} \leftarrow \mathbf{O}_{\text{shard}}^{(i)} \oslash \left(\sqrt{\mathbf{V}_{\text{shard}}^{(i)}} + \varepsilon\right)$

13: $\quad \hat{\eta} = 0.2\eta\sqrt{\text{Numel}(\widehat{\mathbf{O}}_{\text{shard}}^{(i)})}/\|\widehat{\mathbf{O}}_{\text{shard}}^{(i)}\|_F$

14: $\quad \mathbf{W}_{\text{shard}}^{(i)} \leftarrow \mathbf{W}_{\text{shard}}^{(i)} - \eta\lambda\mathbf{W}_{\text{shard}}^{(i)} - \hat{\eta}\widehat{\mathbf{O}}_{\text{shard}}^{(i)}$

15: **end for**

---

## 3.3 Overhead Analysis

**Memory Overhead.** NorMuon maintains Muon's memory efficiency. For a weight matrix $W \in \mathbb{R}^{m \times n}$, the memory consumption of optimizer states for each optimizer is: (1) Adam: $2mn$ (first and second moment estimates); (2) Muon: $mn$ (first moment only); (3) NorMuon: $m(n+1)$ (first moment + per-neuron second-moment statistics). The additional memory overhead of NorMuon compared to Muon is negligible (1/n factor), while remaining 50% more memory-efficient than Adam.

**Communication Overhead.** NorMuon introduces moderate additional communication compared to standard FSDP training. Under FP32 training with AdamW, the per-parameter communication cost is: 4 bytes (forward all-gather) + 4 bytes (backward all-gather) + 4 bytes (gradient reduce-scatter) = 12 bytes. With NS5 iteration computed in BF16 precision, NorMuon requires: 12 bytes (standard FSDP communication) + 2 bytes (momentum gather, BF16) + 2 bytes (update scatter, BF16) = 16 bytes.

This represents a 33% increase in communication volume. When parameters use BF16, the relative overhead increases to 50%. However, this communication can be overlapped with orthogonalization computation to minimize latency

impact. We see communication overlap as an implementation optimization and hence omit the details in Algorithm 2 for simplicity. Please refer to Appendix C for an illustration of the pipeline. In our experiments (Section 4.1.5), we demonstrate that the per-iteration latency of NorMuon is only 3% higher than AdamW, while achieving significantly better convergence efficiency.

# 4 Experiments

In this section, we conduct pretraining experiments across four different model scales to validate the effectiveness of NorMuon: 124M, 350M, 1.1B, and 5.4B parameters. For the larger models (1.1B and 5.4B), we adopt the experimental setup from previous work on architecture scaling (Ren et al., 2025), with results and configurations presented in Section 4.1. For the smaller models (124M and 350M), we follow the experimental setting of Modded-NanoGPT (Jordan et al., 2024a), with results and settings provided in Section 4.2. We include extensive ablation studies that justify our design choices, along with detailed efficiency analyses (Section 4.1.4 and 4.1.5).

## 4.1 Experiments on 1.1B and 5.4B Models

### 4.1.1 SETUP

**Models.** We follow a simple linear rule from prior works (Kaplan et al., 2020; Ren et al., 2025) for scaling the architectural shape. Specifically, for a model with depth $d$ layers, we configure the architecture as follows: hidden dimension $\alpha d$, number of attention query heads $d$, number of attention key-value heads $d/4$, and MLP intermediate dimension $4\alpha d$, where $\alpha = 128$. The $\alpha$ and ratios are derived relative to Llama-3-8B (Grattafiori et al., 2024). Our 1.1B and 5.4B parameter models correspond to depths of $d = 16$ and $d = 28$ layers, respectively.

**Dataset.** We conduct pretraining on the SlimPajama dataset (Soboleva et al., 2023). The 1.1B and 5.4B models are trained on 50B and 100B tokens, respectively—corresponding to approximately $2\times$ and $1\times$ the Chinchilla token budget. Additional results for the 1.1B model trained at the $1\times$ Chinchilla setting (20B tokens) are provided in Appendix A.2.

**Hyperparameters.** We first tune the hyperparameters on the 1.1B model as base, then we employ Depth-$\mu$p (Yang et al., 2023) to scale the learning rate inversely proportional to $\sqrt{d}$ based on model depth. The base learning rate is set to $4 \times 10^{-4}$ with a base model depth of 16. The learning rate schedule consists of a 1B token warmup phase followed by linear decay to 0. We apply 0.1 weight decay for 2D parameters in hidden layers and zero weight decay for others to enhance training stability (Ren et al., 2025). The batch size is fixed at 2M tokens with a sequence length of 4096

tokens. For optimization, we use the following configurations: Adam optimizer with $(\beta_1, \beta_2) = (0.9, 0.95)$, Muon optimizer with $\beta_1 = 0.95$ following (Jordan et al., 2024b), and our proposed NorMuon optimizer with $(\beta_1, \beta_2) = (0.95, 0.95)$. More details on the hyperparameters tuning are provided in Appendix A.1

**Baselines.** We compare NorMuon against three established optimizers: AdamW (Loshchilov and Hutter, 2017), the standard adaptive optimizer with decoupled weight decay; Muon (Jordan et al., 2024b), which applies orthogonalization to update directions; and Dion (Ahn et al., 2025), a scalable orthogonalization-based method that uses low-rank power iteration. For all orthogonalization-based optimizers (including NorMuon), we apply orthogonalization to the 2D weight matrices in the hidden layers, covering attention projections (QKV and output projections) and MLP up/down projections. Following standard practice (Jordan et al., 2024b), embeddings, layer norms, biases, and the LM head are trained with AdamW.

*Table 1.* Efficiency Gain over AdamW. Calculated as percentage reduction in training steps required to reach the same final loss achieved by AdamW. Dion's performance on 5.4B model is excluded due to resource constraints and similar performance with Muon on 1.1B scale.

| Optimizer | 1.1B Model (%) | 5.4B Model (%) |
|---|---|---|
| Muon | 10.43 | 5.60 |
| Dion | 10.43 | – |
| NorMuon | 21.74 | 13.17 |

### 4.1.2 MAIN RESULTS

Figure 2 presents the validation loss curve across different model scales. NorMuon demonstrates consistent and substantial improvements over all baseline optimizers. While orthogonalization-based optimizers (Muon and Dion) already outperform AdamW, NorMuon amplifies this advantage through the integration of our proposed neuron-wise adaptive learning rate.

To quantify the performance gains, Table 1 reports the percentage reduction in training steps required for each optimizer to achieve the same final validation loss as AdamW. NorMuon achieves the best efficiency gains of 21.74% and 13.17% for the 1.1B and 5.4B models, respectively.

### 4.1.3 BENCHMARK PERFORMANCE.

We further evaluate the 1.1B and 5.4B pretrained models using standard downstream evaluation tasks for pretrained checkpoints, including OpenBookQA (Mihaylov et al., 2018), HellaSwag (Zellers et al., 2019), ARC-Easy (Clark et al., 2018), WSC273 (Levesque et al., 2012), Winogrande (Sakaguchi et al., 2020), BoolQ (Clark et al., 2019), and PIQA (Bisk et al., 2020) As shown in Table 2,

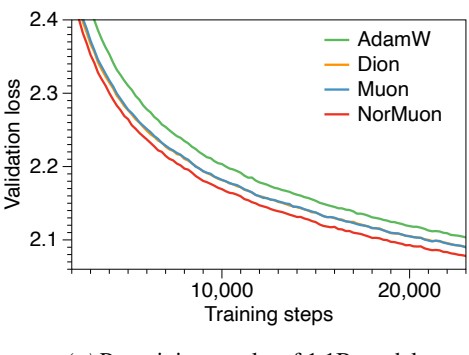
(a) Pretraining results of 1.1B model.

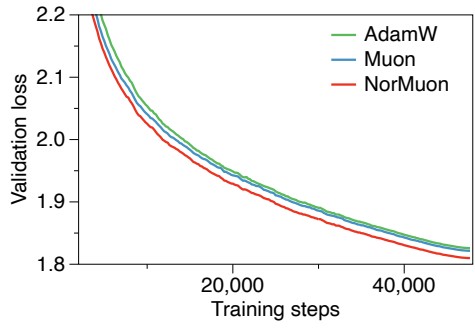
(b) Pretraining results of 5.4B model.

Figure 2. Comparison of different optimizers on pretraining on 1.1B (a) and 5.4B (b) Transformers. NorMuon outperforms other baselines by notable margin.

across both model sizes, NorMuon achieves the best average downstream performance and yields the best results on almost all tasks.

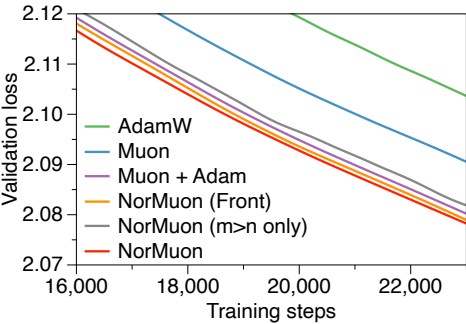

Figure 3. Ablation studies of NorMuon on 1.1B pretraining experiments, including the default post-NS5 NorMuon, NorMuon (Front) with pre-NS5 neuron-wise normalization, Muon+Adam, and selective normalization variants.

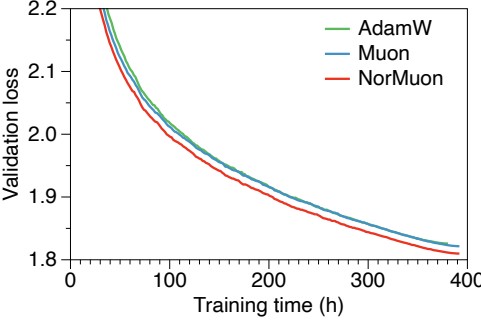

Figure 4. Validation loss vs wallclock time on 5.4B pretraining experiments.

### 4.1.4 ABLATION STUDIES.

To validate our design choices in NorMuon, we conduct ablation studies that examine three key aspects: the granularity of adaptive learning rates and the positioning of nor-

malization relative to orthogonalization, and the impact of applying normalization universally versus selectively based on matrix dimensions. The results on pretraining 1.1B model are presented in Figure 3.

**Adaptive Learning Rate Granularity**. We compare our neuron-wise adaptive approach against a coordinate-wise adaptive variant, "Muon+Adam", which applies Adam-style coordinate-wise normalization after Muon's orthogonalization. This approach is related to concurrent work (Si et al., 2025), though we remove the sign-stabilization component because it performed better without it in our setting. Figure 3 shows that Muon+Adam improves over vanilla Muon but remains below NorMuon in the 1.1B pretraining trajectory. We further conduct a controlled tuning study in the 124M Modded-NanoGPT setting, reported in Appendix A.4. In our tested settings, NorMuon slightly outperforms Muon+Adam at its best-tuned configuration, while requiring only per-row second-moment statistics rather than a full per-parameter second-moment tensor. Thus, NorMuon preserves most of Muon's optimizer-state memory advantage over AdamW, whereas Muon+Adam has AdamW-level optimizer-state memory cost, as shown in Section 4.1.5.

**Normalization Positioning**. We further investigate whether the position of neuron-wise normalization relative to Newton-Schulz orthogonalization matters. Besides our default post-NS5 NorMuon, we evaluate "NorMuon (Front)", which applies neuron-wise adaptation before NS5. This variant directly tests the pre-orthogonalization design choice. As shown in Figure 3, NorMuon (Front) still improves over Muon, demonstrating that neuron-wise adaptive normalization is beneficial even when applied before orthogonalization. However, it slightly underperforms the default post-NS5 NorMuon, suggesting that directly balancing the final orthogonalized update is the more effective ordering in our experiments.

**Universal vs. Selective Normalization**: To test whether

*Table 2.* Downstream benchmark performance of 1.1B and 5.4B pretrained models.

| Model | Optimizer | **Avg.** | OpenBookQA | HellaSwag | ARC-E | WSC273 | Winogrande | BoolQ | PIQA |
|---|---|---|---|---|---|---|---|---|---|
| 1.1B | AdamW | 53.80 | 31.00 | 47.35 | 53.45 | 62.64 | 53.12 | 59.14 | 69.91 |
| | Muon | 53.91 | 31.60 | 48.12 | 55.22 | 58.24 | 52.57 | **60.76** | 70.89 |
| | NorMuon | **55.34** | **33.40** | **48.70** | **55.85** | **65.57** | **53.43** | 59.51 | **70.95** |
| 5.4B | AdamW | 60.59 | 35.60 | 59.98 | 62.79 | **72.16** | 58.48 | 61.62 | 73.50 |
| | Muon | 60.65 | 35.40 | 59.99 | 63.51 | 72.10 | 58.33 | 60.73 | 74.48 |
| | NorMuon | **61.74** | **36.40** | **61.35** | **63.93** | **72.16** | **60.30** | **63.46** | **74.59** |

normalization is beneficial to $m \leq n$ matrices as discussed in Section 3.1, we evaluate "NorMuon ($m > n$ only)", which applies normalization only to $m > n$ matrices. We can see that this selective approach underperforms the full NorMuon, demonstrating that applying normalization to those with $m \leq n$ is helpful.

### 4.1.5 COMPUTATIONAL AND MEMORY OVERHEAD ANALYSIS

In this section, we show that overheads incurred by Nor-Muon are manageable and do not diminish the practical benefits.

**Wall-Clock Performance**. Figure 4 presents the validation loss as a function of wall-clock training time. Despite the additional computation required for orthogonalization and neuron-wise normalization, NorMuon maintains substantial performance advantages over AdamW.

**Memory and Computational Overhead**. Table 3 provides a breakdown of the computational and memory requirements for each optimizer. NorMuon achieves comparable memory efficiency to Muon with nearly a 50% reduction compared to AdamW or Muon + Adam. In terms of computational cost, NorMuon introduces only a 2.9% increase in training step time compared to AdamW. We can observe that forward and backward passes dominate the training step time, making the cost of optimizer step relatively small. NorMuon's neuron-wise norm computation adds minimal cost relative to orthogonalization operations.

We further profile the step time of NorMuon under two ablated configurations: (i) without communication overlap and (ii) without our orthogonalization distribution strategy. As shown in Table 3, communication overlap provides a clear reduction in optimizer-step latency, while distributing the orthogonalization workload across GPUs is critical for keeping overhead low. Removing this distribution increases the optimizer-step time by approximately $2.7\times$. Importantly, this distributed orthogonalization strategy can be and has been applied to Muon as well, ensuring a fair comparison between the two methods.

### 4.2 Experiments on Modded-NanoGPT

To further verify the advantages of NorMuon over Muon, we conduct experiments using Muon's original experimental setting on Modded-NanoGPT (Jordan et al., 2024a).

**Models.** The model architecture is consistent with GPT-2 (Radford et al., 2019), with 124M and 350M parameter configurations obtained by adjusting width and depth.

**Dataset.** We train all models on the FineWeb dataset (Penedo et al., 2024). The 124M model is trained on approximately 3.2B tokens, while the 350M model uses approximately 4B tokens. We also include results on C4 dataset (Raffel et al., 2020) in Appendix A.3.

**Hyperparameters.** Since Muon has already performed extensive hyperparameter tuning in this setting (Jordan et al., 2024b), we adopt their optimized configurations except for $\beta_1$, which we slightly tune. We use a batch size of 512, sequence length of 1024, and the Warmup-Stable-Decay (WSD) learning rate schedule. Training iterations are set to 6,200 for the 124M model and 7,500 for the 350M model. For the 124M model, Adam's parameters uses a learning rate of $3.6 \times 10^{-3}$ with momentum parameters $(\beta_1, \beta_2) = (0.9, 0.95)$. For Muon and NorMuon, we set the learning rate to $3.6 \times 10^{-4}$ and conduct a grid search over $\beta_1 \in \{0.9, 0.95\}$, reporting the best result. For NorMuon, $\beta_2$ is set to 0.95. For the 350M model, Adam employs differentiated learning rates: 0.3 for the embedding layer and $3 \times 10^{-3}$ for the output layer, with momentum parameters $(\beta_1, \beta_2) = (0.8, 0.95)$. For hidden layers, Muon and NorMuon use a learning rate of $7.5 \times 10^{-4}$, with $\beta_1$ selected from $\{0.9, 0.95\}$ based on validation performance.

**Results**. Figure 5 presents the comparison between Nor-Muon and Muon on 124M and 350M parameter models. NorMuon consistently outperforms Muon across both model sizes. Since Muon's improvements over Adam have been extensively demonstrated in Jordan et al. (2024b), we focus the figure on the comparison between Muon and Nor-Muon.

To quantify the computational benefits of NorMuon, we conduct an additional analysis where Muon is trained with

*Table 3.* Computational and memory overhead comparison for different optimizers when training a 5.4B parameter model. Training step time includes forward pass, backward pass, and optimizer step. Percentages indicate relative increase compared to AdamW baseline.

| Optimizer | Memory cost of optimizer states (GB) | Optimizer step time (s) | Training step time (s) |
|---|---|---|---|
| AdamW | 40.56 | 0.02 | 28.73 |
| Muon | 21.14 | 0.83 | 29.56 (2.8%↑) |
| Muon + Adam | 40.56 | 0.85 | 29.58 (3.0%↑) |
| NorMuon | 21.19 | 0.84 | 29.57 (2.9%↑) |
| w/o communication overlap | - | 1.31 | 30.04 (4.5%↑) |
| w/o orthogonalization distribution | - | 2.29 | 31.04 (8.1%↑) |

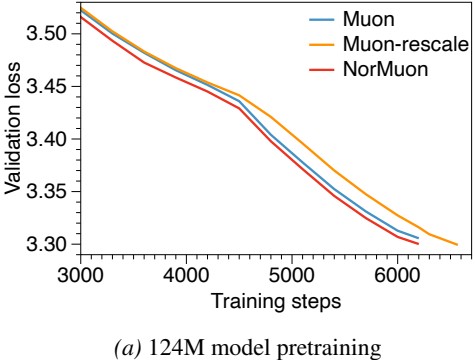

*(a)* 124M model pretraining

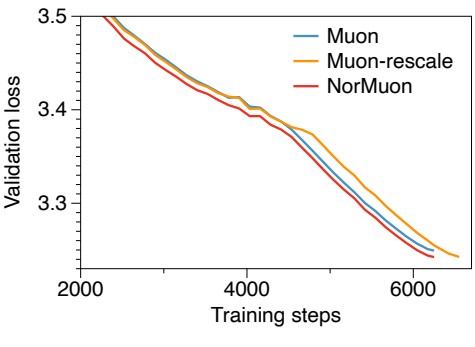

*(b)* 350M model pretraining

*Figure 5.* Comparison of Muon and NorMuon on pretraining 124M (a) and 350M (b) Modded-NanoGPT on FineWeb. NorMuon outperforms Muon by notable margin.

the same learning rate schedule but for a longer total number of iterations, until it reaches the same validation loss as NorMuon (denoted as "Muon-rescale" in Figure 5). More specifically, we sweep total Muon training steps and report the smallest step count whose validation loss is no higher than NorMuon's final validation loss. We observe that on the 124M model, Muon requires 6% more iterations than NorMuon to achieve the same validation loss. On the 350M model, this efficiency gap increases substantially to 15%, demonstrating the advantage of NorMuon over Muon.

**Comparison with normalization-based optimizers**. We additionally compare NorMuon with two normalization-based optimizers, SWAN (Ma et al., 2025) and SCALE (Glentis et al., 2025). For both baselines, we conduct extensive hyperparameter sweeps, with batch size the same as main experiments to ensure fairness. As shown in Figure 6, both SWAN and SCALE notably underperform Muon and NorMuon in our setting. One possible reason is that their original experiments use a much shorter sequence length, whereas the Modded-NanoGPT setting uses sequence length 1024. These results suggest that, under this longer-context pretraining setup, NorMuon's combination of matrix-level orthogonalization and neuron-wise normalization remains more effective than normalization/whitening-only baselines.

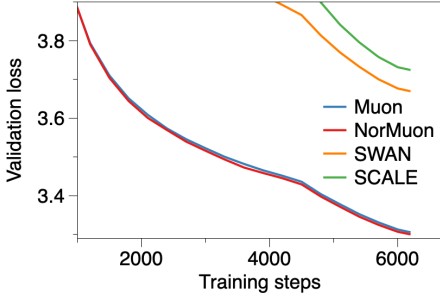

*Figure 6.* Comparison with SWAN and SCALE on 124M Modded-NanoGPT.

## 5  Conclusion

In this work, we introduced **NorMuon**, a simple yet effective optimizer that integrates Muon's orthogonalization with neuron-wise adaptive learning rates. To make NorMuon practical for large-scale training, we developed an efficient distributed implementation under the FSDP2 framework, carefully orchestrating momentum gathering and orthogonalization to eliminate redundant computation and communication overhead. Our experiment results show notable improvement over Muon, demonstrating that orthogonalization and adaptive scaling need not be mutually exclusive; rather, their combination can lead to superior optimization dynamics.

## Impact Statement

This paper presents work whose goal is to advance the field of Machine Learning. There are many potential societal consequences of our work, none which we feel must be specifically highlighted here.

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

# A    Additional Experiments

## A.1    Hyperparameters tuning of main experiments.

In this subsection, we describe how the base hyperparameters were selected for the main experiments and the additional tuning performed to evaluate robustness.

For the 1.1B model, we directly adopt the hyperparameters from TinyLlama (Zhang et al., 2024) for the AdamW baseline, as these settings are already well-tuned. To further verify this, we conduct an additional hyperparameter sweep under the 1× Chinchilla token budget, and the results (shown below) confirm that the adopted configuration is near-optimal.
For the Muon baseline, we tune only the learning rate while keeping all other hyperparameters fixed, except for $\beta_1$, which we set to 0.95 following the original Muon blog. For NorMuon, to avoid any implicit tuning bias, we use exactly the same hyperparameters as Muon and vary only the learning-rate ratio.

To ensure a rigorous comparison across optimizers, we perform a controlled hyperparameter sweep on the 1.1B model trained with a 1× Chinchilla budget (20B tokens). For each optimizer, we sweep over: learning rate (for AdamW) or learning-rate ratio (for Muon/NorMuon), EMA coefficients $\beta_1$, $\beta_2$, and weight decay. For Muon and NorMuon, we tune the learning-rate ratio (Algorithm 1)—not the raw learning rate—consistent with prior work (Liu et al., 2025a). The results are summarized below; the top two configurations for each optimizer are bolded. The first row in each block corresponds to the configuration used in the main experiments.

*Table 4.* Hyperparameters Sweep of AdamW

| Learning Rate | $\beta_1$ | $\beta_2$ | WD | Val. Loss |
|---|---|---|---|---|
| 4e-4 | 0.9 | 0.95 | 0.1 | **2.176** |
| 2e-4 | – | – | – | 2.242 |
| 8e-4 | – | – | – | 2.182 |
| – | 0.8 | – | – | 2.203 |
| – | 0.95 | – | – | 2.179 |
| – | – | 0.9 | – | 2.176 |
| – | – | 0.98 | – | 2.190 |
| – | – | – | 0 | 2.213 |
| – | – | – | 0.2 | **2.174** |

*Table 5.* Hyperparameters Sweep of Muon

| Learning Rate Ratio | $\beta_1$ | WD | Val. Loss |
|---|---|---|---|
| 0.2 | 0.95 | 0.1 | **2.134** |
| 0.1 | – | – | 2.154 |
| 0.4 | – | – | **2.135** |
| – | 0.9 | – | 2.175 |
| – | 0.98 | – | 2.145 |
| – | – | 0 | 2.189 |
| – | – | 0.2 | 2.144 |

Across all sweeps, additional tuning yields minimal improvements, confirming that the current configurations were already well-chosen. Importantly, the best NorMuon configuration consistently achieves lower validation loss than the best Muon and AdamW settings, further supporting its robustness.

## A.2    Experiments of 1.1B model trained on 20B tokens.

Figure 7 presents the validation loss curve 1.1B model trained on 20B tokens. NorMuon demonstrates consistent improve-

*Table 6.* Hyperparameters Sweep of NorMuon

| Learning Rate Ratio | $\beta_1$ | $\beta_2$ | WD | Val. Loss |
|:---:|:---:|:---:|:---:|:---:|
| 0.2 | 0.95 | 0.95 | 0.1 | **2.128** |
| 0.1 | – | – | – | 2.144 |
| 0.4 | – | – | – | **2.128** |
| – | 0.9 | – | – | 2.167 |
| – | 0.98 | – | – | 2.135 |
| – | – | 0.9 | – | 2.128 |
| – | – | 0.98 | – | 2.134 |
| – | – | – | 0 | 2.176 |
| – | – | – | 0.2 | 2.134 |

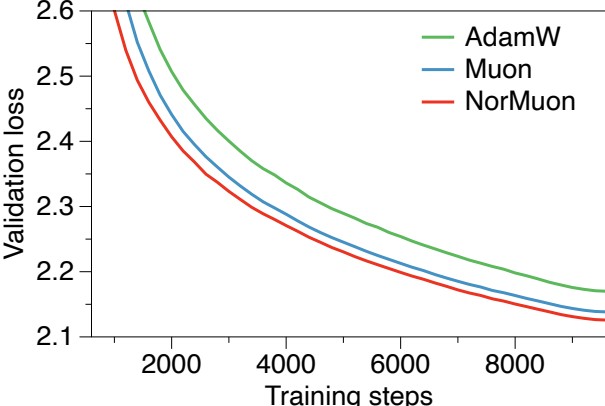

*Figure 7.* Pretraining results of 1.1B model on 20B tokens.

ments. NorMuon achieves the best efficiency gains of 25.04%, outperforming Muon by 5.88%.

### A.3 Experiments on Modded-NanoGPT with C4 dataset

The original Modded-NanoGPT leaderboard adopt the FineWeb dataset. To further demonstrate the robustness and generality of NorMuon, we replicate the benchmark using a different large-scale corpus, the C4 dataset (Raffel et al., 2020). Following the 1×Chinchilla token budget, we train a 124M model on approximately 3B tokens and a 350M model on 7B tokens, and we re-tune the hyperparameters of both Muon and NorMuon for this new setting.

As shown in Figure 8, NorMuon consistently achieves lower validation loss than Muon across the entire training trajectory for both model sizes. When we extend Muon's training long enough to match NorMuon's final validation loss, Muon requires approximately 6% more optimization steps. These findings mirror the trends observed on FineWeb, further supporting that NorMuon provides more sample-efficient training across datasets and scales.

### A.4 Additional tuning study for Muon+Adam

We further compare NorMuon with a coordinate-wise adaptive variant, denoted as Muon+Adam, under the 124M Modded-NanoGPT FineWeb setting. Muon+Adam first applies Muon's Newton-Schulz orthogonalization and then applies Adam-style coordinate-wise second-moment normalization to the orthogonalized update. This variant therefore combines Muon's matrix-level orthogonalization with full coordinate-wise adaptivity, but it requires maintaining a full per-parameter second-moment tensor.

Table 7 reports a comparable sweep over learning rate and momentum coefficients. The first row of each block corresponds to the default configuration, and "–" indicates that the corresponding hyperparameter is kept fixed at the default value for that block. Under this sweep, NorMuon achieves a slightly better best validation loss than Muon+Adam. At the same

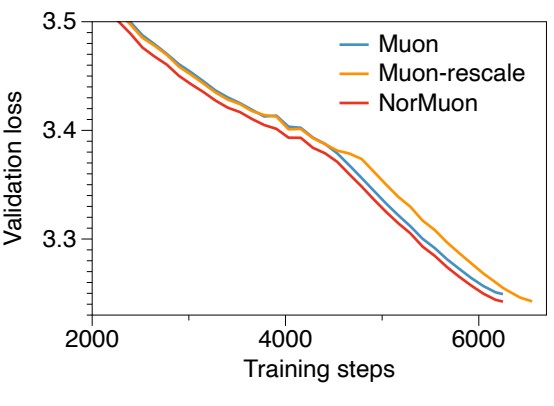

*(a)* Pretraining results of 124M model.

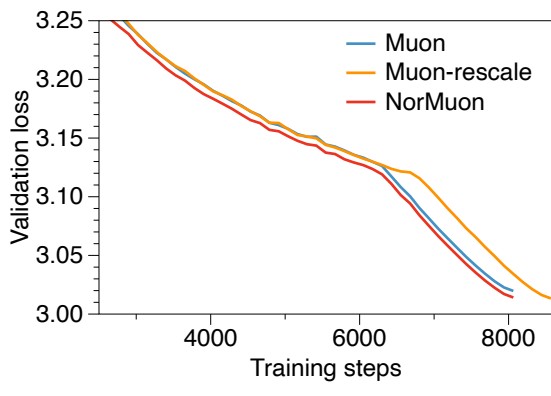

*(b)* Pretraining results of 350M model.

*Figure 8.* Comparison of Muon and NorMuon on pretraining 124M (a) and 350M (b) Modded-NanoGPT on C4. NorMuon shows consistently lower validation loss throughout training.

time, Muon+Adam appears relatively robust to learning-rate tuning, possibly due to the stabilizing effect of coordinate-wise scaling. Overall, these results support the claim: in the settings we tested, NorMuon matches or slightly outperforms Muon+Adam at its best-tuned configuration, while using substantially less optimizer state.

*Table 7.* Tuning comparison between NorMuon and Muon+Adam on 124M Modded-NanoGPT pretraining on FineWeb. The first row of each block is the default configuration, and "–" means the corresponding hyperparameter is held fixed at the default value for that block.

| Optimizer | Learning Rate | $\beta_1$ | $\beta_2$ | Val. Loss |
|---|---|---|---|---|
| NorMuon | $3.6 \times 10^{-4}$ | 0.95 | 0.95 | **3.2996** |
| | $1.8 \times 10^{-4}$ | – | – | 3.3250 |
| | $7.2 \times 10^{-4}$ | – | – | 3.3100 |
| | $1.44 \times 10^{-3}$ | – | – | 3.7352 |
| | – | 0.85 | – | 3.3106 |
| | – | 0.90 | – | 3.3003 |
| | – | – | 0.90 | 3.3024 |
| | – | – | 0.99 | 3.3003 |
| Muon+Adam | $3.6 \times 10^{-4}$ | 0.95 | 0.99 | **3.3011** |
| | $1.8 \times 10^{-4}$ | – | – | 3.3217 |
| | $7.2 \times 10^{-4}$ | – | – | 3.3047 |
| | $1.44 \times 10^{-3}$ | – | – | 3.4339 |
| | – | 0.85 | – | 3.3180 |
| | – | 0.90 | – | 3.3052 |
| | – | – | 0.90 | 3.3085 |
| | – | – | 0.95 | 3.3063 |

### A.5 Ablation Experiments on Modded-NanoGPT

To further verify the effectiveness of NorMuon, we conducted several ablation experiments under the setting of training a 350M Modded-NanoGPT on FineWeb, and show the results in Figure 9:

**(1)** Standard NorMuon, denoted as "NorMuon" in Figure 9.

**(2)** Standard Muon used in original setting of Modded-NanoGPT (Jordan et al., 2024b), denoted as "Muon" in Figure 9.

**(3)** Applying normalization directly to Muon's update such that the update is strictly $\sqrt{m \times n}$, denoted as "Muon w/ normalization" in Figure 9.

**(4)** applying NorMuon only to weight matrices with $m > n$, while using the normalized muon mentioned in (3) for all other weight matrices, denoted as "NorMuon ablation" in Figure 9.

We can see that although Muon with normalization performs slightly better than Muon in the early stages, it is eventually surpassed by Muon, indicating that the effectiveness of NorMuon cannot be attributed to normalization. Furthermore, since weight matrices with $m > n$ correspond only to the MLP up-projection matrices, which constitute only a small portion of the model, applying NorMuon only to this subset of parameters greatly diminishes the effect of NorMuon, resulting in only a marginal improvement over Muon.

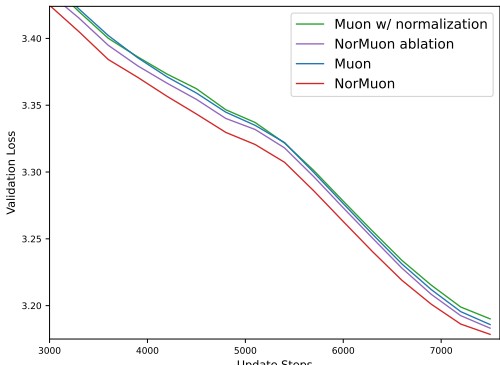

*Figure 9.* Ablation studies on pretraining 350M model.

### A.6   Supervised Finetuning Experiments.

To further validate the effectiveness of NorMuon beyond pretraining, we additionally evaluate it in a supervised fine-tuning (SFT) setting. We perform long-reasoning mathematical SFT using Qwen 2.5 Math 7B (Yang et al., 2024) as the base model. The model is fine-tuned on the Mixture-of-Thoughts (Hugging Face, 2025) dataset, where we use a subset of 87k prompts covering mathematics and science. Following prior work, we assess downstream performance on MATH 500 (Lightman et al., 2023), AIME 2024, AIME 2025, and GPQA (Rein et al., 2024).

| Optimizer | Avg | MATH 500 | AIME24 | AIME25 | GPQA |
|---|---|---|---|---|---|
| AdamW | 43.36 | 80.60 | 25.73 | 23.96 | **43.18** |
| Muon | 43.58 | 81.20 | **27.50** | 23.49 | 42.11 |
| NorMuon | **44.02** | **82.20** | 26.60 | **24.69** | 42.61 |

*Table 8.* Supervised fine-tuning evaluation on mathematical reasoning benchmarks.

As shown in Table 8, NorMuon attains the highest average score and achieves the best performance on most individual tasks. These results demonstrate that NorMuon remains effective in the SFT regime, not just in pretraining. We also note that prior work (Liu et al., 2025a) highlights that the alignment between pretraining and fine-tuning optimizers can notably influence downstream performance. Therefore, the benefits of NorMuon may be even more pronounced when applied to LLMs that were pretrained with Muon-like optimizers.

### A.7   Step time with 32 A100 GPUs.

On a 32-GPU configuration, we measure the per-step optimizer time and end-to-end training step time for both Adam and NorMuon. The results are summarized in Table 9. As shown, NorMuon incurs only a 2.9% overhead in total training step time relative to Adam, despite a larger per-optimizer computation. This minimal slowdown arises from two factors: (i) the forward and backward passes dominate overall wall-clock time; and (ii) as GPU count increases, our distributed orthogonalization scheme parallelizes across devices, reducing per-GPU optimizer cost.

*Table 9.* Overhead on 32 GPUs. NorMuon adds only 2.9% overhead in total step time.

| Model | Optimizer Step Time (s) | Training Step Time (s) |
|---|---|---|
| Adam | 0.016 | 14.99 |
| NorMuon | 0.460 | 15.43  (+2.9%) |

## B  Implementation Details

For experiments involving 1.1B and 5.4B parameter models, we conducted training on 2 nodes, each equipped with 8 A100 GPUs (80GB) connected via NVLink for optimized inter-GPU communication. Training duration was approximately 2 days for the 1.1B model and 7 days for the 5.4B model.

All experiments using Modded-NanoGPT, including both 124M and 350M parameter models, were performed on a single node with 8 A100 GPUs.

## C  Communication Overlap Timeline Visualization

To aid understanding of our distributed optimizer design, we provide a timeline visualization (Fig. 10) illustrating the inter-group pipelining used in NorMuon. For clarity, we depict a simplified example with three parameter groups and two GPUs. We group parameters and (i) launch an asynchronous gather for group 1; (ii) once group 1 is ready, launch the gather for group 2 while performing orthogonalization on group 1; (iii) iterate this process over all groups. The gather on different GPUs for different parameters can be achieved by efficient all-to-all operation. This figure represents the idealized scheduling pattern; in practice, the exact boundaries of the communication and computation blocks may vary depending on parameter sizes, communication bandwidth, and hardware characteristics.

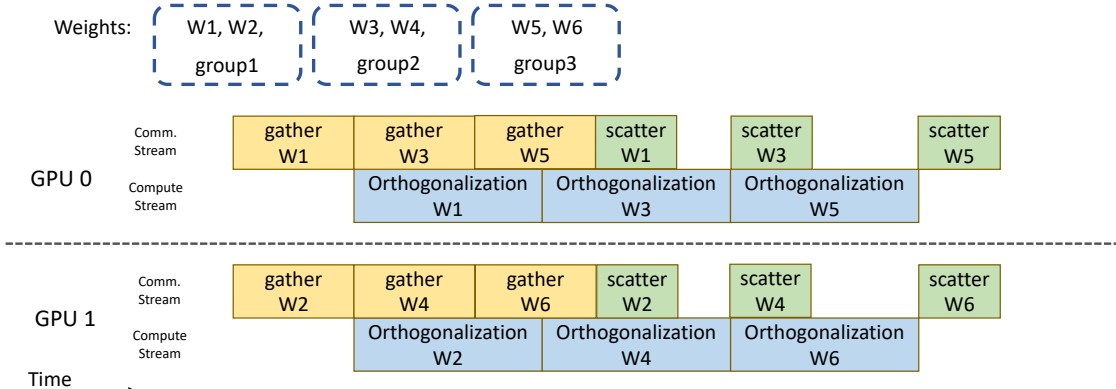

*Figure 10.* Timeline visualization of the distributed NorMuon optimizer, showing how gather for group $k+1$ overlaps with orthogonalization of group $k$ across separate CUDA streams.

## D  Visualization of Updates' singular value and neuron norm.

Figure 11, figure 12, and figure 13 present the visualization of updates' singular value and neuron norms similar to figure 1 on different training stages and layers. We also provide more details on what are plotted. For each optimizer, we plot the pre-learning-rate update $\mathbf{U}_t$: SGD w/ momentum uses $\mathbf{U}_t = \mathbf{M}_t$; AdamW uses $\mathbf{U}_t = \mathbf{M}_t^{\mathrm{Adam}}/(\sqrt{\mathbf{V}_t^{\mathrm{Adam}}} + \epsilon)$; Muon uses $\mathbf{U}_t = \mathbf{O}_t = \mathrm{NS5}(\mathbf{M}_t)$; and NorMuon uses $\mathbf{U}_t = \widehat{\mathbf{O}}_t = \mathbf{O}_t/(\sqrt{\mathrm{ExpandRows}(\mathbf{v}_t)} + \epsilon)$, with divisions taken elementwise. Both singular values and row $L^2$ norms are normalized by their respective means.

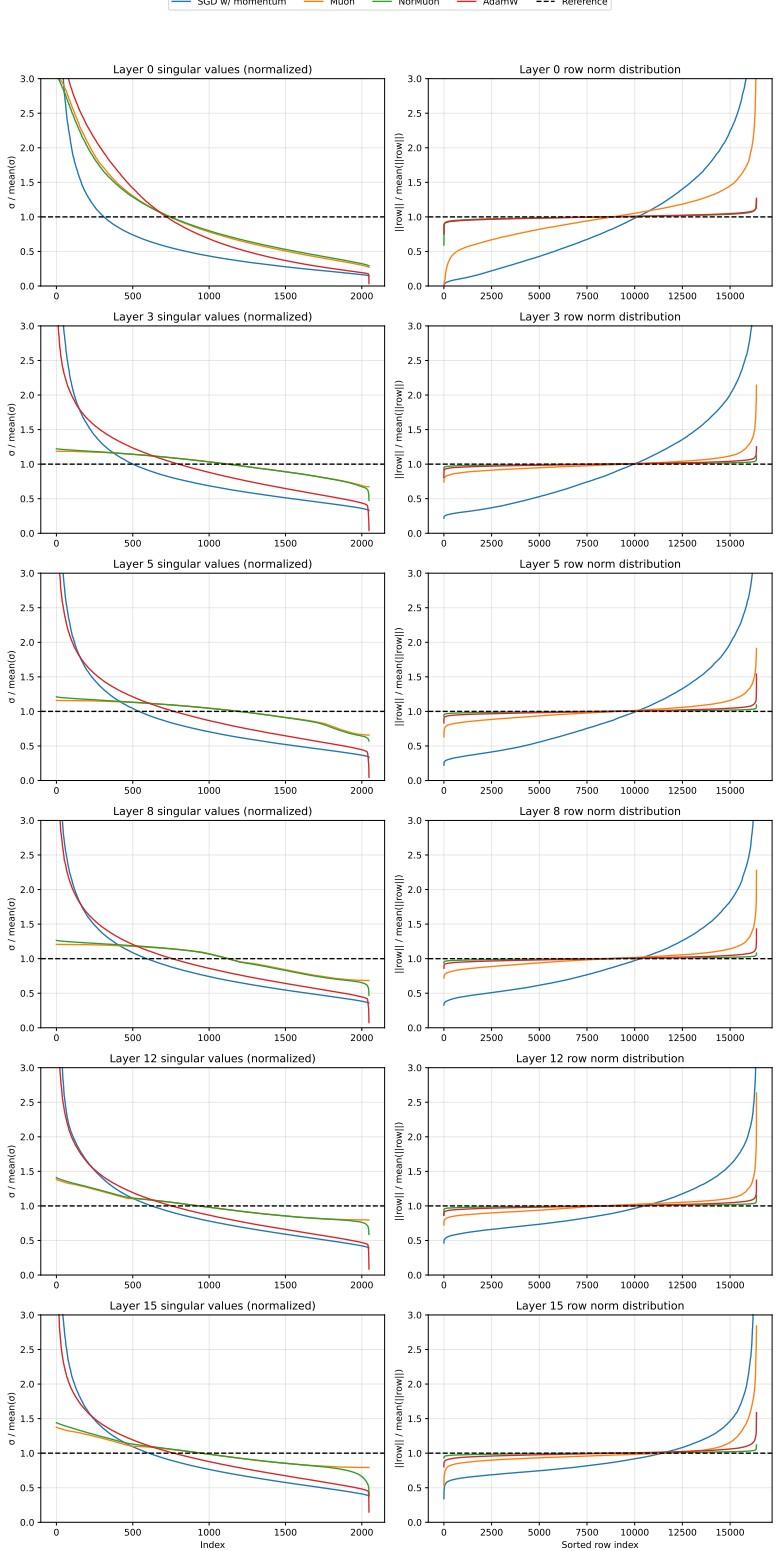

*Figure 11.* Visualization of optimization geometry on different layers on 1.1B model pretraining early checkpoints (10B).

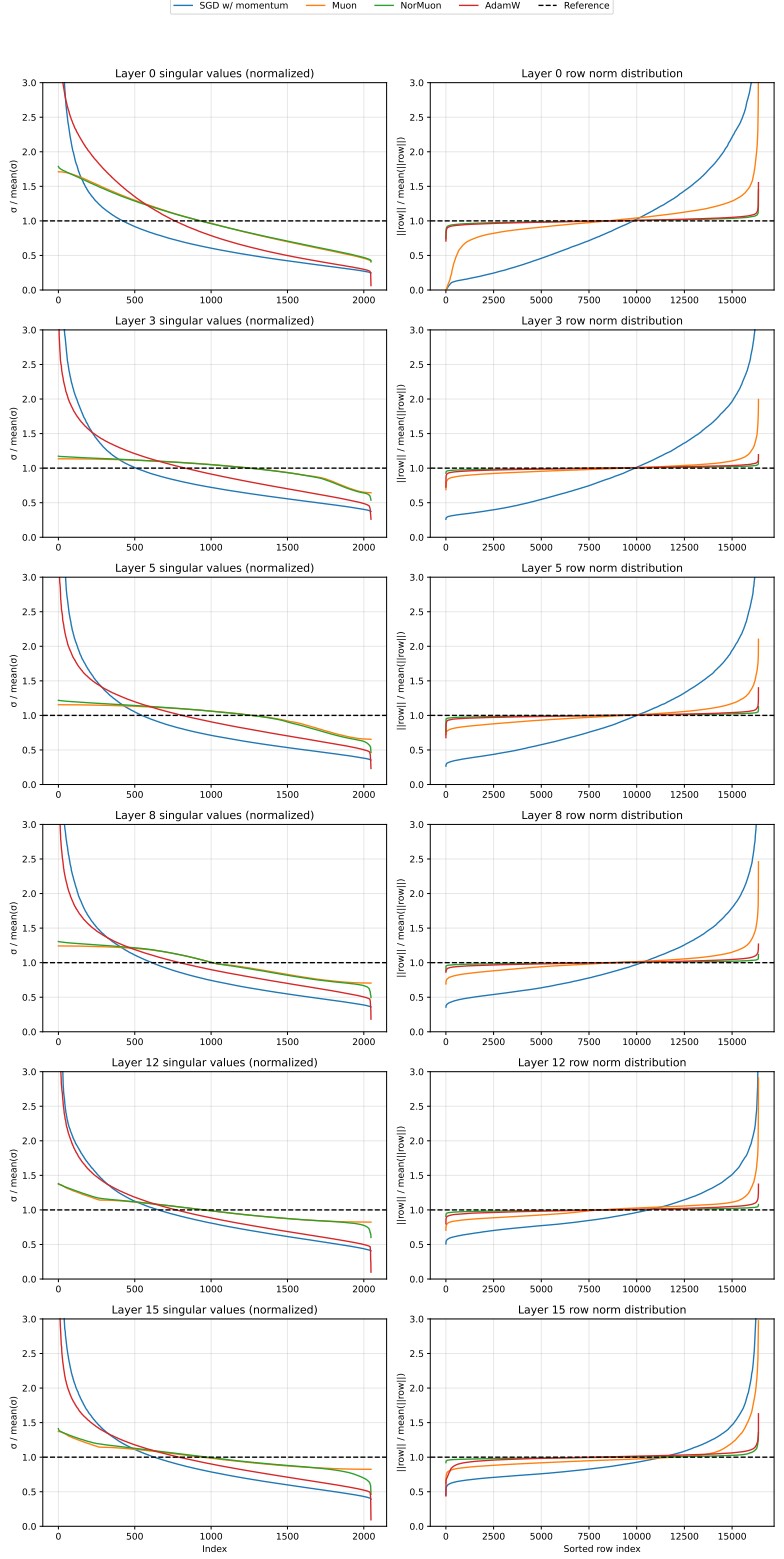

*Figure 12.* Visualization of optimization geometry on different layers on 1.1B model pretraining middle checkpoints (30B).

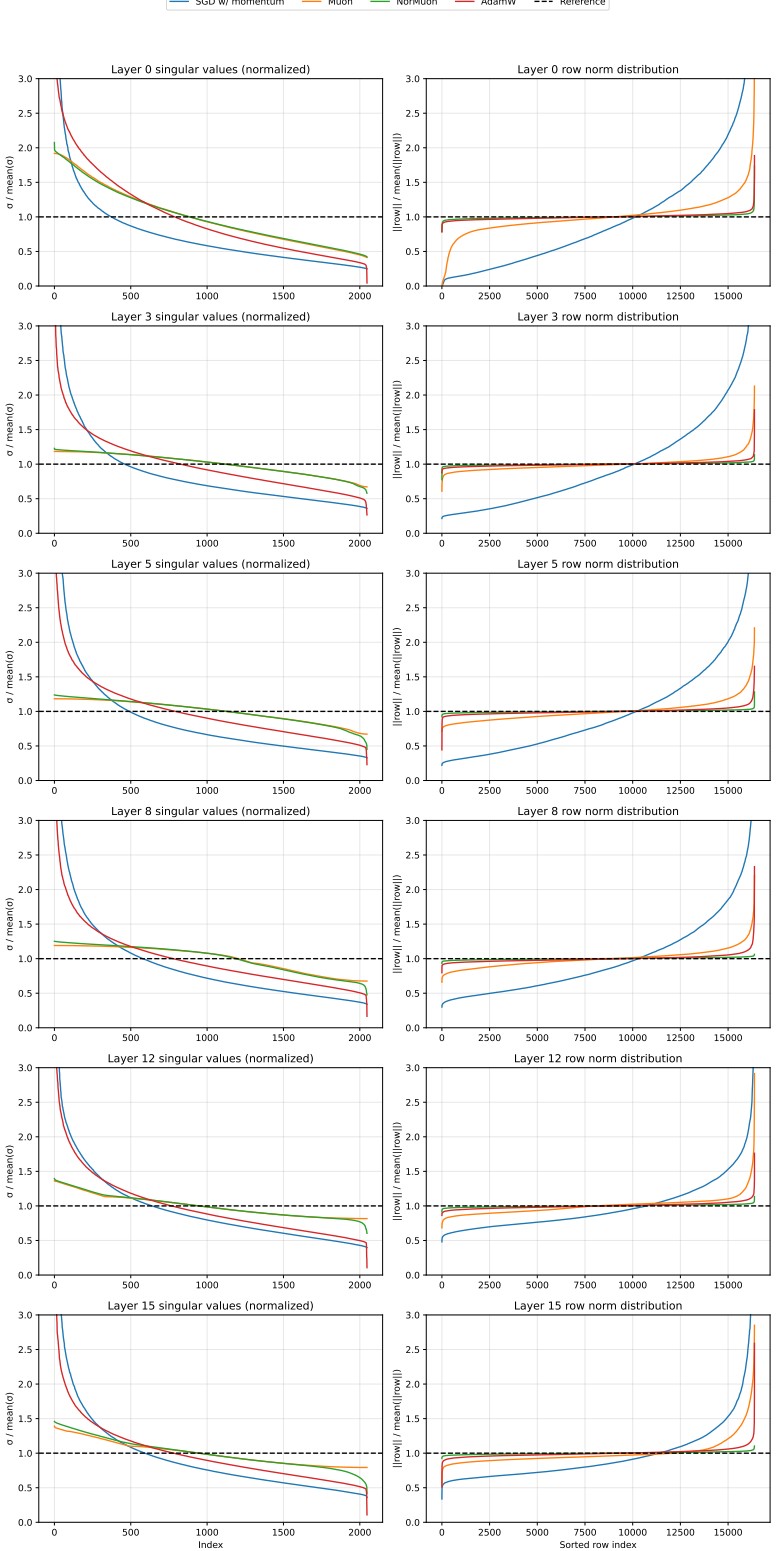

*Figure 13.* Visualization of optimization geometry on different layers on 1.1B model pretraining final checkpoints (50B).

