# OpenReview forum: "NorMuon: Making Muon more efficient and scalable"
_ICML.cc/2026/Conference — ICML 2026 spotlight_

### Official Review · Reviewer_vXbR · 2026-02-24

**Soundness:** 3
**Presentation:** 2
**Significance:** 3
**Originality:** 2
**Overall Recommendation:** 4
**Confidence:** 4

**Summary:**

This paper introduces NorMuon, an optimizer that combines Muon's orthogonalization-based updates with neuron-wise adaptive learning rates. The key observation motivating this work is that while Muon's Newton-Schulz iterations improve matrix conditioning by orthogonalizing momentum, they produce updates with highly variable per-neuron norms, leading to imbalanced parameter utilization. NorMuon addresses this by maintaining per-neuron variance statistics (exponential moving averages of squared orthogonalized updates) and applying row-wise normalization after orthogonalization. The paper also presents a distributed implementation under FSDP2 that strategically distributes orthogonalization computations across GPUs to avoid redundancy. Experiments on language model pretraining (124M to 5.4B parameters) demonstrate that NorMuon achieves better training efficiency than AdamW and Muon, while maintaining comparable memory usage to Muon.

**Compliance With Llm Reviewing Policy:**

Affirmed.

**Key Questions For Authors:**

1. Terminology clarification: You describe $v_t$ as "second-order momentum" throughout the paper, but this appears to be a second moment estimate, not a second-order (curvature-based) method. Could you explain why you mention second-order optimizers in Section 2.1?

2. Sparsity and pruning trade-offs: Your method enforces "balanced parameter utilization" to prevent neuron domination. However, natural sparsity often emerges precisely when some neurons receive negligible updates. Does NorMuon inhibit the formation of sparse representations? Have you tested whether models trained with NorMuon are more difficult to prune (e.g., via magnitude-based pruning) compared to Muon or AdamW? Should practitioners favor quantization over pruning when deploying NorMuon-trained models?

3. Hyperparameter robustness: The learning rate scaling factor of 0.2 is adopted from prior work. Given that NorMuon changes the update structure significantly, did you experiment with different scaling factors specifically for this method? Is the 0.2 factor still optimal?

**Limitations:**

1. Terminology error: The paper describes $v_t$ as "second-order momentum," though this refers to second-moment estimation rather than second-order optimization. While this distinction is often blurred in informal usage, clarifying that NorMuon uses second-moment statistics would help readers accurately situate the method within the adaptive optimization landscape and avoid confusion with Hessian-based approaches.

2. Sparsity-pruning trade-off: The method's core innovation, balanced parameter utilization, likely comes at the cost of reduced natural sparsity. This may prohibit efficient deployment via pruning techniques that rely on magnitude disparities between neurons.

3. Narrow architectural scope: The evaluation focuses exclusively on decoder-only Transformer language models. The effectiveness of NorMuon on CNNs, GNNs, or mixture-of-experts architectures remains unverified.

4. Theoretical understanding: The method relies on empirical observations about neuron norms without theoretical characterization of how this affects convergence rates or the loss landscape geometry.

**Strengths And Weaknesses:**

Strengths:

1. Well-motivated design: The analysis in Figure 1 clearly illustrates the complementary issues: Muon has more uniform singular value distributions but creates high variance in per-neuron norms, while AdamW achieves uniform neuron norms but poor conditioning. This motivates the combination effectively.

2. Practical distributed implementation: The FSDP2-compatible algorithm (Algorithm 2) addresses real scalability challenges by distributing orthogonalization computation and overlapping communication with computation.

3. Comprehensive empirical validation: The evaluation spans four model scales (124M, 350M, 1.1B, 5.4B), two datasets (FineWeb and SlimPajama), and includes supervised fine-tuning results, demonstrating broad applicability.

4. Memory efficiency: The method has ~50% memory savings over AdamW, adding negligible overhead.




Weaknesses:

1. Terminology confusion: The authors repeatedly refer to $v_t$ as "second-order momentum" (e.g., Abstract, Section 3.1, Algorithm 1), but this is a second moment estimate, not a second-order method. True second-order optimization uses Hessian or Fisher Information Matrix curvature (as in K-FAC/Shampoo, which the authors correctly identify elsewhere). This conflation suggests imprecise understanding of optimization theory and misrepresents the method's computational complexity (linear like Adam, not cubic like second-order methods).

2. Sparsity implications unexplored: The paper explicitly targets "balanced parameter utilization" to prevent certain neurons from dominating. However, this normalization likely suppresses the natural emergence of sparsity patterns (where some neurons become inactive or "dead"). The authors do not discuss whether this uniform activation pattern affects post-hoc pruning strategies and sparsity-based acceleration techniques.

3. Limited theoretical depth: While the empirical motivation is strong, the paper lacks theoretical analysis of convergence properties or generalization bounds under the proposed update rule.

---

> ### Author Rebuttal · Authors · 2026-03-31
>
> Thank you for the valuable comments! Please see our response as follows:
>
> > **W1/Q1 The authors repeatedly refer to as "second-order momentum", but this is a second moment estimate, not a second-order method. Could you explain why you mention second-order optimizers in Section 2.1?**
>
> We thank the reviewer for pointing this out. We agree that “second-order momentum” is imprecise terminology here. Our intended meaning was Adam’s exponential moving average of squared gradients, i.e., the second moment estimate. Our original phrasing followed prior optimizer work [1]. We will revise the wording throughout the paper to “second moment estimate” for clarity, and emphasize that our method retains Adam-like first-order computational complexity rather than that of true second-order optimizers.
>
> Relatedly, our discussion of Shampoo/K-FAC/SOAP in Section 2.1 is intended purely as background and positioning for the broader optimizer landscape, not as a claim that NorMuon belongs to the same computational class.
>
> > **W2/Q2 The authors do not discuss whether the uniform activation pattern affects post-hoc pruning strategies and sparsity-based acceleration techniques. Should practitioners favor quantization over pruning when deploying NorMuon-trained models?**
>
> The interaction with post-hoc compression is an interesting deployment question, but it is outside the scope of the current paper. Our focus is to improve training efficiency and optimization quality for a fixed model size, rather than to optimize for pruneability. For this reason, we do not want to claim, without direct evidence, that NorMuon either helps or hurts post-hoc pruning. Designing optimizers that explicitly make models easier to prune would be an interesting research direction in its own right, rather than a standard objective of optimizer design.
>
> More broadly, balancing update scales does not by itself imply incompatibility with later compression. Adaptive optimizers such as Adam also rescale updates across coordinates or blocks, yet this is conceptually distinct from whether the resulting model can be effectively pruned afterward; many pruning studies start from Adam-trained models [2].
> The choice of quantization or pruning for NorMuon-trained models is highly dependent on the deployment setting and downstream task, and a proper comparison of pruning-versus-quantization trade-offs would require a dedicated study. We will add this as a limitation and leave a systematic sparsity/compression analysis to future work.
>
> > **W3. Limited theoretical depth**
>
> We acknowledge that we do not provide a complete theoretical analysis of convergence. This is largely due to the complexity of modern LLM training, where end-to-end optimization dynamics remain beyond the reach of current theory. This limitation is shared by many recent optimizer works, including Muon and SOAP, which are likewise motivated by geometric or second-order intuition together with empirical evidence rather than full guarantees. In the same spirit, our contribution here is primarily empirical: we identify a deficiency in Muon’s neuron-level update behavior, provide intuition for why it may hinder optimization, and show that a neuron-wise adaptive mechanism effectively addresses it in practice. We leave a deeper theoretical treatment to future work, and refer the reviewer to our response to Reviewer yCE4 W1 for additional intuition and a toy example illustrating why NorMuon helps.
>
> > **Q3. Hyperparameter robustness of the learning rate scaling factor 0.2**
>
> We adopt the 0.2 scaling factor from prior work [3], but it is not fixed without validation. Varying this factor is largely equivalent to varying the effective learning-rate, which we do tune. As shown in Table 6, the 1.1B sweep yields the same best validation loss for 0.2 and 0.4, suggesting that NorMuon is reasonably robust to this choice.
>
>
> > **L3. Narrow architectural scope**
>
> Our work is intentionally scoped to LLM pretraining, as stated in the abstract.
> For this reason, we evaluate NorMuon only on decoder-only Transformer language models, which are the primary large-scale setting where optimizer memory efficiency, conditioning, and distributed overhead are major practical bottlenecks. Evaluating other architectures would be interesting, but would constitute a separate empirical study rather than a necessary component of the current paper. Consistent with this, recent optimizer studies including SOAP [4], GaLore [5] also restrict their experiments to transformer-based architectures.
>
> **Reference**
>
> [1] Adam-mini: Use Fewer Learning Rates To Gain More
>
> [2] LLM Pruning and Distillation in Practice: The Minitron Approach
>
> [3] Muon is Scalable for LLM Training
>
> [4] SOAP: Improving and Stabilizing Shampoo using Adam
>
> [5] GaLore: Memory-Efficient LLM Training by Gradient Low-Rank Projection

---

> > ### Author Rebuttal · Reviewer_vXbR · 2026-04-03
> >
> > I appreciate the authors' clarifications. I will maintain my original decision.

---

### Official Review · Reviewer_r2ax · 2026-02-25

**Soundness:** 2
**Presentation:** 2
**Significance:** 2
**Originality:** 2
**Overall Recommendation:** 4
**Confidence:** 5

**Summary:**

This paper proposes NorMuon, an optimizer that improves upon Muon by addressing the issue of non-uniform per-neuron update norms. NorMuon adds neuron-wise adaptive learning rates through lightweight second-order momentum statistics and row-wise normalization applied after orthogonalization. The method preserves Muon's conditioning benefits while ensuring more balanced parameter updates. The authors also develop an efficient distributed implementation that minimizes computational overhead while strategically distributing orthogonalization across devices.

**Compliance With Llm Reviewing Policy:**

Affirmed.

**Final Justification:**

Most concerns solved

**Key Questions For Authors:**

1. Why is normalization applied after orthogonalization rather than before? What is the theoretical or empirical justification for this ordering? Have you tested the alternative ordering?

2. For the distributed implementation, can you provide more details on: (a) how communication is overlapped, (b) per-layer profiling to validate the claimed efficiency, (c) how the approach generalizes to different FSDP configurations beyond row-wise sharding?

3. The paper mentions applying orthogonalization to concatenated QKV matrices. Can you clarify how these matrices are concatenated? Given that NorMuon uses row-wise normalization, this detail is important for reproducibility.

**Limitations:**

See weaknesses and questions.

**Strengths And Weaknesses:**

**Strengths**

1. Clear problem identification: The paper effectively diagnoses that Muon's orthogonalization improves conditioning but leads to imbalanced per-neuron update norms, providing a concrete motivation for the proposed solution.

2. Simple and practical design: NorMuon's addition of row-wise normalization after orthogonalization is elegant and easy to implement, with minimal memory overhead (O(n) per row).

3. Rigorous ablation studies: The paper systematically validates design choices including neuron-wise vs. coordinate-wise adaptation, normalization positioning (before vs. after orthogonalization), and the impact of different components.

4. Distributed implementation: The FSDP2-based distributed algorithm is well-designed to minimize communication overhead while maintaining efficiency.

**Weaknesses**

1. Limited evaluation scope: The tasks described in Section 4.1.3 cannot be regarded as standard pretraining tasks; rather, they should be considered SFT tasks. In addition, the types of models evaluated in the paper are relatively limited, and incorporating experimental results on more modern (7B+) models such as LLaMA3, Qwen3, and Gemma3 would help to better assess the contribution of the method.

2. There is a lack of comparison with other related normalize-based optimizers, such as SWAN [1] and SCALE [2].

3. Single dataset per scale: The experiments use only one dataset per model scale (SlimPajama for 1.1B/5.4B, FineWeb for 124M/350M). Testing across multiple datasets would improve generalizability and robustness claims.

4. Theoretical justification: The motivation for why more uniform per-neuron update norms lead to better performance is not rigorously established. While the empirical results are promising, a deeper theoretical understanding of the relationship between update norm uniformity and optimization dynamics would strengthen the contribution.

5. Hyperparameter tuning fairness: The paper uses different hyperparameters (learning rates, betas, weight decay) across optimizers. To ensure fair comparison, it should be clarified whether equal tuning effort was applied to all baselines, or whether the reported improvements could be attributed to better hyperparameter selection for NorMuon.

6. Design choice justification: Several design choices need better justification. For example, why is the normalization applied after orthogonalization rather than before? Sensitivity analysis for these hyperparameters would be valuable.

[1] Glentis, Athanasios, et al. "A minimalist optimizer design for LLM pretraining." arXiv preprint arXiv:2506.16659 (2025).

[2] Ma, Chao, et al. "Swan: Sgd with normalization and whitening enables stateless llm training." arXiv preprint arXiv:2412.13148 (2024).

---

> ### Author Rebuttal · Authors · 2026-03-31
>
> Thank you for the insightful comments. Please see our response below:
> > **W1. The tasks tested cannot be regarded as standard pretraining tasks. The types of models evaluated are relatively limited.**
>
> Thank you for this helpful suggestion. We agree that the wording in Section 4.1.3 is imprecise: these are not standard pretraining tasks, but downstream tasks of pretrained checkpoints [1,2]. We will revise the wording accordingly.
>
> We also agree that broader architectural coverage would strengthen the paper, and we view this as valuable future work.
> We would like to clarify that our current evaluation is already relatively broad in scale, covering four model sizes from 124M to 5.4B parameters.  We note that recent optimizer studies in this area typically evaluate on one primary architecture family [1,3,4,5]. We therefore view the current evidence as supporting the effectiveness of NorMuon on the tested settings.
>
> > **W2. There is a lack of comparison with SWAN and SCALE.**
>
> Thank you for highlighting the relevant baselines. To address this concern, we added comparisons against both methods, where we also conduct hyperparameter sweeps.
> In the 124M pretraining, both methods perform worse than Muon and NorMuon, as shown in Figure [link](https://anonymous.4open.science/r/NorMuon_ICML26_rebuttal-F9F8/swan_scale_124M.png). We would like to note that both papers use a very short sequence length (256), which may be the reason why they perform worse under our setting with a more standard sequence length 1K. We will include these results and the discussion in the revision.
>
> > **W3. Single dataset per scale.**
>
> We additionally include a second dataset experiment in Appendix A.3: for the 124M/350M setting, we replicate the comparison on C4 and observe the same qualitative advantage of NorMuon.
>
> > **W4. Theoretical justification for uniform per-neuron update norms.**
>
> We acknowledge that we do not provide a complete theoretical analysis. This is largely due to the complexity of modern LLM training, where end-to-end optimization dynamics remain beyond the reach of current theory. This limitation is shared by many recent optimizer works, including Muon and SOAP [5], which are likewise motivated by geometric or second-order intuition together with empirical evidence rather than full guarantees. In the same spirit, our contribution here is primarily empirical: we identify a deficiency in Muon’s neuron-level update behavior, and show that a neuron-wise adaptive mechanism effectively addresses it in practice. We leave a deeper theory to future work, and refer the reviewer to our response to Reviewer yCE4 W1 for additional intuition and a toy example illustrating this effect.
>
> > **W5. Hyperparameter tuning.**
>
> We used the same tuning effort for all baselines and for NorMuon. Appendix A.1 reports controlled sweeps for AdamW, Muon, and NorMuon over learning rate, momentum coefficients, and weight decay under the same protocol.
>
> > **W6/Q1. Design choice justification**
>
> Our goal is to correct the neuron-level imbalance that remains after Muon’s orthogonalization. For this reason, post-orthogonalization normalization is the natural design: it preserves Muon’s geometric preconditioning and then rescales neuron-wise step sizes. Pre-normalization, in contrast, modifies the quantity that is subsequently orthogonalized. We also test this alternative directly in Figure 3 via NorMuon (Front) and find that it underperforms the standard design.
>
> > **Q2. Distributed implementation details.**
>
> (a) We provide the overlap strategy in Appendix D. In brief, we pipeline communication and computation across parameter groups: while the gather for group k+1 is launched asynchronously, orthogonalization for group k proceeds on a separate stream.
>
> (b) Our current paper reports end-to-end efficiency profiling in Table 3. In particular, we break down optimizer-step time and total training-step time, and further include ablations without communication overlap and without orthogonalization distribution. These results isolate the contribution of the main system components and support the claimed efficiency.
>
> (c) We discussed the sharding setting in line 233. NorMuon is not tied to FSDP2 row-sharding. Under more general sharding schemes, the same row-wise normalization can still be applied by aggregating per-row statistics with a lightweight all-reduce. The additional communication relative to Muon is very small, since it involves only per-row scalars rather than full parameter tensors.
>
> > **Q3. Details on concatenated QKV matrices.**
>
> The Q, K, and V weights are concatenated along the output dimension before applying orthogonalization. We will clarify this in the revision.
>
> [1] Fantastic Pretraining Optimizers and Where to Find Them
>
> [2] Qwen 2.5 Technical report
>
> [3] A minimalist optimizer design for LLM pretraining.
>
> [4] Swan: Sgd with normalization and whitening enables stateless llm training.
>
> [5] SOAP: Improving and Stabilizing Shampoo using Adam

---

> > ### Author Rebuttal · Reviewer_r2ax · 2026-04-02
> >
> > Dear Authors,
> >
> > Thank you for the time that you have taken to respond to my initial review. In light of your responses, I have adjusted my score accordingly.

---

### Official Review · Reviewer_yCE4 · 2026-03-12

**Soundness:** 3
**Presentation:** 3
**Significance:** 3
**Originality:** 2
**Overall Recommendation:** 5
**Confidence:** 4

**Summary:**

This paper introduces a new neural network optimizer. The authors empirically show that, in contrast to momentum SGD, both Muon and Adam stabilize update directions in different ways (singular values and neuron-wise norms). They then introduce NorMuon, a method that combines both types of stabilization by integrating Muon and Adam-mini. Furthermore, the paper develops an efficient distributed implementation and discusses learning-rate scaling with model size. Lastly, the new method is compared with prior methods and different variants of the proposed method on LLMs with up to 5.4B parameters.

**Compliance With Llm Reviewing Policy:**

Affirmed.

**Key Questions For Authors:**

1. Can you provide any mathematical or numerical explanation of why NorMuon performs better beyond the conditioning of the updates (see Strengths and Weaknesses)?
2. Do you have any intuition or numerical results explaining why Muon + Adam performs worse than NorMuon? How does it behave with respect to optimization geometry (as in Figure 1)?
3. You tested NorMuon (Front) vs. NorMuon; did you also try this with Muon + Adam?

**Limitations:**

yes

**Strengths And Weaknesses:**

The paper describes an interesting empirical phenomenon regarding the conditioning of the update directions of different optimizers and, based on this observation, introduces a new method that combines the purported advantages of Adam and Muon. The resulting method appears to work very well in practice. The experimental section is quite thorough and includes many interesting ablations, such as comparisons with Muon + full Adam and investigations of when to apply normalization in NorMuon. The paper is well written and presents a clear and easy-to-follow narrative.

The main weakness is that, while the method works well in practice, it is unclear why the normalization improves performance. This may not be fully understood even for previous methods such as Adam or Muon, but it would have been helpful to include at least some empirical comparisons aimed at developing intuition for why NorMuon works so well. The authors seem to imply that improving the conditioning of the update matrices leads to better performance. Is this claim supported by any mathematical or empirical evidence? The authors state regarding Adam that “This coordinate-wise preconditioning improves stability and convergence in heterogeneous settings,” so one possible experiment would be to check whether NorMuon is more stable than Muon.

---

> ### Author Rebuttal · Authors · 2026-03-31
>
> Thank you for the insightful comments and positive review! Please find our response below:
> > **W1/Q1 While the method works well in practice, it is unclear why the normalization improves performance.**
>
> Thank you for highlighting the need for stronger intuition on why NorMuon helps. Our intuition is that Muon and NorMuon improve different levels of update geometry. Muon primarily acts at the matrix level, flattening the singular-value spectrum, but it does not guarantee balanced update magnitudes across neurons, as shown in Figure 1. NorMuon preserves the favorable spectrum while also making neuron norms much more uniform.
>
> A useful interpretation is through the neuron’s functional step size: for row update $\Delta w_i$ of neuron $i$ and input $h$, the pre-activation change $\Delta z_i=\langle \Delta w_i, h\rangle$ satisfies $\mathbb{E}[\Delta z_i^2]=\Delta w_i^\top \Sigma_h \Delta w_i$. Under isotropic input covariance, this is roughly proportional to $\\|\Delta w_i\\|_2^2$, so row norm serves as a proxy for how much that neuron’s feature actually moves. Large row-norm imbalance therefore means that only a subset of neurons makes most of the representation progress. Our interpretation is that NorMuon improves optimization by correcting this residual neuron-level imbalance after orthogonalization. This is also consistent with prior work: [1] suggests that balancing feature updates across neurons can improve training efficiency, and [2] supports the view that the appropriate granularity for adaptivity is often block-wise rather than fully coordinate-wise.
>
> To further illustrate the mechanism, we construct a simple isotropic teacher-student regression toy with a tall linear layer $W\in\mathbb{R}^{256\times 8}$ and loss $L(W)=\frac{1}{2}\\|W-U\\|\_F^2$.
>
> Let $O\in\mathbb R^{m\times n}$ be the exact Muon update (the exact polar factor), and let $r\_i=\\|O_{i:}\\|\_2$ denote the norm of row $i$.
> We define the controlled family
>
> $\widetilde O^{(\alpha)}=c_\alpha D\_\alpha O, D\_\alpha = \mathrm{diag}(r\_1^{-\alpha},\dots,r_m^{-\alpha}), c_\alpha = \frac{\\|O\\|\_F}{\\|D\_\alpha O\\|_F}.$
>
> Equivalently, row $i$ is rescaled as
> $\widetilde O^{(\alpha)}\_{i:} = c\_\alpha r\_i^{-\alpha} O\_{i:}. $
>
> Thus, $\alpha=0$ recovers Muon, $\alpha=1$ equalizes row norms (NorMuon-like row balancing), and $\alpha<0$ makes the update more imbalanced.
>
> We then analyze this family. As shown in Figure [linkC](https://anonymous.4open.science/r/NorMuon_ICML26_rebuttal-F9F8/toy_experiment_plots/alpha_first_step_imbalance_metrics.png), increasing $\alpha$ steadily reduces row-norm imbalance. And the update remains near-isometric, with condition number changing only mildly (Figure [linkD](https://anonymous.4open.science/r/NorMuon_ICML26_rebuttal-F9F8/toy_experiment_plots/alpha_first_step_condition_number.png)). Correspondingly, the first-step loss reduction improves with increasing $\alpha$ (Figure [linkE](https://anonymous.4open.science/r/NorMuon_ICML26_rebuttal-F9F8/toy_experiment_plots/alpha_first_step_loss_drop.png)). This benefit also persists in final loss (Figure [linkF](https://anonymous.4open.science/r/NorMuon_ICML26_rebuttal-F9F8/toy_experiment_plots/alpha_final_loss.png)). These results support our interpretation that, even when matrix conditioning is already good, reducing neuron-level imbalance can further improve optimization. The experiment code is included [here](https://anonymous.4open.science/r/NorMuon_ICML26_rebuttal-F9F8/toy_experiment.ipynb).
>
>
> > **Q2 Do you have any intuition explaining why Muon + Adam performs worse than NorMuon? How does it behave in Figure 1?**
>
> We add a geometry plot in the same setting as Figure 1 for Muon+Adam [here](https://anonymous.4open.science/r/NorMuon_ICML26_rebuttal-F9F8/Muon_Adam_figure1.png). Muon+Adam produces neuron-wise norms that are similarly uniform to NorMuon, but it has a slightly worse singular spectrum. This suggests that coordinate-wise normalization can equalize neuron magnitudes, but does so in a way that partially corrupts the matrix structure after orthogonalization. NorMuon achieves a better tradeoff. In the toy example, Muon+Adam update also shows a worse condition number than NorMuon (Fig. [linkG](https://anonymous.4open.science/r/NorMuon_ICML26_rebuttal-F9F8/toy_experiment_plots/named_first_step_geometry.png)).
>
> > **Q3 You tested NorMuon (Front) vs. NorMuon; did you also try this with Muon + Adam?**
>
> Yes. In our early exploration, we also tested a Muon+Adam (Front) variant. Empirically, this variant performed worse than Muon, so we did not include it in the ablation figure. Our hypothesis is that this ordering is less compatible: coordinate-wise normalization rescales entries independently, while the subsequent orthogonalization reshapes the update at the matrix level, which can partially interfere with the intended effect.
>
> [1] Rotational Equilibrium: How Weight Decay Balances Learning Across Neural Networks
>
> [2] Adam-mini: Use Fewer Learning Rates To Gain More

---

> > ### Author Rebuttal · Reviewer_yCE4 · 2026-04-04
> >
> > Thanks for this detailed rebuttal, it answers all my concerns. I will keep my initial score.

---

### Official Review · Reviewer_fGpc · 2026-03-13

**Soundness:** 3
**Presentation:** 3
**Significance:** 3
**Originality:** 3
**Overall Recommendation:** 5
**Confidence:** 3

**Summary:**

This paper proposes NorMuon, an optimizer that combines Muon’s matrix-level orthogonalization of momentum updates with "neuron-wise" adaptive learning rates inspired by Adam-mini. In addition, the paper presents a distributed NorMuon implementation built on FSDP2, distributing orthogonalization computation across devices and leveraging row-wise sharding to compute row-normalization locally with negligible extra communication. The authors provide an overhead analysis (memory / computation / communication) and report wall-clock overheads that remain small relative to overall step time. Empirically, the paper evaluates pretraining across multiple scales (Modded-NanoGPT 124M/350M and Llama-like 1.1B/5.4B), with extensive ablations and downstream evaluations; NorMuon consistently outperforms AdamW, Muon, and Dion and also shows gains in a supervised fine-tuning setup.

**Compliance With Llm Reviewing Policy:**

Affirmed.

**Final Justification:**

I recommend acceptance. This is a technically solid and practically meaningful paper on optimizer design for large-scale LLM training. NorMuon is a simple but effective combination of Muon-style orthogonalization and neuron-wise adaptive scaling, and the paper supports its claims with strong empirical results across multiple model scales, ablations, downstream evaluation, and distributed implementation analysis. Overall, I view the contribution as significant and likely to have meaningful impact in the optimizer / efficient training subarea.

My main initial concerns were about clarity, tuning fairness, and the justification of several design choices rather than about core soundness. After reading the rebuttal, I think the authors addressed these points well. In particular, they clarified the exact quantities plotted in Figure 1 and stated that the same qualitative trends hold across layers/checkpoints; they explained the rationale for applying normalization after orthogonalization; they committed to improving the FSDP2 explanation and adding the missing citation for the 0.2 factor; and they provided additional evidence on batch-size robustness, weight-decay sensitivity, and the Muon-rescale procedure. They also clarified that code is planned for release.

The point that remained least fully resolved for me was the comparison to Muon+Adam. That said, the follow-up discussion was constructive: the authors appropriately narrowed their claim, provided an additional tuning study, and showed that NorMuon is at least competitive and slightly better at best-tuned performance in the tested small-scale setting, rather than asserting a universal advantage.

Overall, the rebuttal reinforced my positive assessment. While I still think the final paper would benefit from polishing the presentation, expanding related work, and making several implementation/details-oriented clarifications explicit in the main text, I did not see evidence of a fundamental weakness in the method or evaluation. I therefore weigh the strengths more heavily than the remaining weaknesses.

**Key Questions For Authors:**

1. Figure 1 definitions/representativeness: What exactly is the “update matrix” used for the singular value spectrum and per-neuron norm plots (raw momentum $M_t$, orthogonalized $O_t$, normalized $\tilde{O}_t$, or the final scaled step)? Please add explicit formulas. Also, why specifically the 8th-layer MLP up-projection at the middle checkpoint, how consistent is the qualitative picture across layers/checkpoints?

2. Learning-rate scaling constant: The method states that the 0.2 stability factor is adopted from prior work. Which specific paper/blog is this taken from? How sensitive are results to this constant (and/or the learning-rate ratio) at least at small scale?

3. Muon+Adam ablation details: The “Muon+Adam” variant underperforms NorMuon. How were its hyperparameters tuned (LR/ratio, betas, epsilon, WD, any sign-stabilization choices)? Was it given a comparable tuning budget to NorMuon? I would like to understand whether the underperformance is fundamental (interaction between orthogonalization and coordinate-wise adaptivity) or potentially due to tuning constraints.

4. Distributed implementation clarity: Could you provide a more concrete explanation (and possibly pseudocode detail) of the gather/scatter/all-to-all pattern and how parameters are assigned to ranks for orthogonalization under FSDP2? Also, is code for this distributed optimizer implementation available (or planned to be released)?
5. “Muon-rescale” methodology + training setup sensitivity: How exactly is the iteration count determined for “Muon-rescale” (matching NorMuon’s final validation loss), do you use interpolation between checkpoints and is it robust across seeds? Additionally, do you have any evidence (even small-scale) that the reported gains persist under different batch sizes or different WD settings for the Adam-updated parameter subsets?

**Limitations:**

I have not sees the paper discussing limitations. "Negative societal impact" is not very relevant in this case.

**Strengths And Weaknesses:**

### Strengths
- Strong empirical focus on a very practical and timely problem: improving LLM pretraining efficiency. The evaluation spans both smaller “speedrunning” style settings (Modded-NanoGPT) and multi-billion-parameter pretraining, which makes the story more compelling than single-scale results.
- The optimizer construction is simple and plausibly useful in practice: NorMuon keeps Muon’s attractive memory footprint (no full second moment) while adding only per-neuron statistics, which is a clean way to inject adaptivity without reverting to Adam-scale memory.
- The paper contains multiple layers of evidence: main pretraining curves, downstream benchmark evaluation, and ablations on the design choices (normalization placement, granularity of adaptivity, and selectively applying normalization). This helps with soundness and reduces the chance that results are due to a single lucky configuration.
- The distributed FSDP2 implementation is a meaningful engineering contribution. Distributing orthogonalization across ranks (instead of replicating NS5 everywhere) and pipelining communication with orthogonalization are exactly the kinds of details that determine whether such optimizers are viable at scale, and the paper makes an effort to quantify overheads.
- The optimization-geometry analysis (singular value spectrum + per-neuron norms) is a good narrative device and helps explain why Muon + neuron-wise scaling might be complementary rather than redundant.

### Weaknesses
- Figure 1 would benefit from explicit mathematical definitions of what is plotted (and what matrix is being analyzed). The labels suggest “singular value of update (normalized)” and “per-neuron update norm,” but the reader is left guessing whether this is the raw momentum $M_t$, the orthogonalized update $O_t$, the final scaled update after normalization, etc. Adding formulas would make the argument substantially clearer.
- Related work coverage for “Orthogonal Update Methods” feels incomplete historically. The discussion focuses on Muon/Dion and recent work, but misses relevant pre-Muon lines of work on spectral/orthogonal update or matrix-geometry descent (e.g., Spectral Descent-type ideas, Scion). Even a short paragraph acknowledging this prior lineage would strengthen positioning and originality claims.
- Section 3 (Method) contains several design choices that are not obvious a priori: (i) applying NS5 before neuron-wise normalization, (ii) the specific row-statistic used (mean of squared entries across columns), and (iii) the extra global RMS-style rescaling with the 0.2 stability factor. The paper does run ablations, which is great, but it would materially improve readability if Section 3 explicitly pointed to the relevant ablation figures/tables at the moment each choice is introduced.
- The learning-rate scaling constant “0.2 is adopted from prior work” needs a concrete citation (which prior work?) and ideally a brief sensitivity note. This constant appears as a stability knob in the core update rule, so it should not be left uncited.
- The distributed NorMuon section is valuable but hard to parse on a first read, especially for readers not already familiar with FSDP2/sharding semantics. A short “FSDP2 in one paragraph” explanation (row-wise sharding implications, what gather/scatter means here, what is replicated vs sharded) would help a lot, and some appendix implementation details (e.g., the overlap timeline) might be worth moving into the main text in condensed form.
- Experimental setup questions that currently weaken interpretability:
  * Batch size is fixed at 2M tokens for the larger-scale runs. It is unclear whether NorMuon’s gains are stable across batch sizes (even a small-scale ablation would be informative).
  * The weight decay scheme applies WD to 2D hidden-layer matrices but uses zero WD for the rest (which are trained with AdamW). This is plausible for stability, but the paper should justify why no WD for the Adam-updated parameters is the right choice and how sensitive results are to this design.
  * The “Muon-rescale” comparison (training Muon longer until it reaches NorMuon’s final validation loss) is a useful way to quantify step-efficiency, but the paper should state precisely how the matching point is determined.
- Minor presentation issue: in the supervised fine-tuning Table 7, the GPQA column appears to have an incorrect bold highlight (NorMuon’s GPQA is bolded, but AdamW’s GPQA number is higher). This is small, but it should be corrected.

### Significance + originality
Overall, I find the paper strong and potentially high-impact. Optimizer improvements that translate into consistent step-efficiency gains at billion-parameter scale are practically meaningful, and the combination of Muon-style orthogonalization with Adam-mini-style granularity is an appealing, creative synthesis rather than a trivial tweak. The distributed implementation work also increases the chance that the method is adoptable beyond a research prototype. I do think the paper would benefit from polishing and a more complete prior-work framing for orthogonal-update methods, but these are fixable issues rather than fundamental flaws.

---

> ### Author Rebuttal · Authors · 2026-03-31
>
> Thank you for the insightful comments and positive review! Please find our response below:
>
> > **W1/Q1 What is the matrix analyzed in Figure 1? Need formulas. How consistent is the conclusion across layers/checkpoints?**
>
> We provide the precise definitions below and will include these formulas in the paper for clarity. The plotted object is the optimizer-transformed update matrix: for SGD with momentum, the momentum matrix; for AdamW, the variance-normalized momentum update; for Muon, the orthogonalized update $O_t$; and for NorMuon, the row-normalized orthogonalized update $\hat O_t$. Figure 1(a) shows the singular values of this matrix normalized by their mean, and Figure 1(b) shows the sorted row-wise $L_2$ norms normalized by their mean. Because both quantities are mean-normalized, the final global scalar rescaling in NorMuon does not affect the plotted shape; the difference comes from the underlying transformation itself.
>
> The main text uses the 8th-layer MLP up-projection as a representative example, and the same qualitative trend is consistent across layers and checkpoints (Appendix Figures 10–12).
> > **W2 Related work coverage for “Orthogonal Update Methods” feels incomplete historically.**
>
> We agree and will expand this discussion in the revision to acknowledge earlier spectral / geometry-aware optimization work [1,2].
>
> > **W3/W4/Q2 Section 3 contains several design choices that are not obvious a prior: (i) NS5 before neuron-wise normalization, (ii) row-statistics, (iii) 0.2 factor, this constant needs a concrete citation and sensitivity note.**
>
> Thank you for these suggestions. We will revise section 3 accordingly to improve clarity.
>
> (i) We apply NS5 before normalization because NorMuon aims to balance the orthogonalized update across neurons; if normalization is applied first, the subsequent orthogonalization can change that balance. This is also consistent with the ablation in Figure 3, where NorMuon (Front) is slightly worse than the standard design.
>
> (ii) We use row-wise statistics in the same spirit as Adam-mini: the right granularity for adaptivity is often block-wise because neural-network Hessians are often near block-diagonal. In our setting, rows naturally correspond to neurons, so row-wise normalization directly targets neuron-level balance.
>
> (iii) We agree the citation for the 0.2 factor should be added explicitly. This factor is inherited from [3].  Changing this factor is largely equivalent to changing the effective learning rate. We will clarify this and add a sensitivity note; Table 6 shows that the method is reasonably robust to this factor, with 0.2 and 0.4 giving the same best validation loss
>
>
> > **W5/Q4 A short “FSDP2 in one paragraph” explanation will improve readability. Is the code planned to be released?**
>
> We will add a short paragraph for FSDP2 and move the overlap details to the main text as suggested.
>
> Concretely, under FSDP2, each 2D parameter matrix is sharded row-wise along dim 0 across GPUs. Thus, the parameter, gradient, optimizer states are all stored as local row shards, not fully replicated on every GPU. In our setting, “replicated computation” refers to the naive distributed Muon strategy where multiple ranks reconstruct the same full matrix and run the same NS5 orthogonalization redundantly. Our implementation avoids this. Due to space limit, we do not include the detailed explanation of FSDP concepts here, but we will include them in the revision.
>
> Regarding availability, yes, we plan to release the code.
>
> > **W6/Q5  Do you have any evidence that the reported gains persist under different batch sizes or WD settings? How exactly is the iteration count determined for “Muon-rescale”?**
>
> We include additional results to address these questions.
> For batch size, we ran additional experiments on the 20B-token 1.1B setting with 1M and 4M token batch sizes. We adjusted the learning rate linearly with batch size, and the results (Fig. [linkA](https://anonymous.4open.science/r/NorMuon_ICML26_rebuttal-F9F8/Batchsize_NorMuon.png)) show the same qualitative improvement across these settings.
>
> For weight decay, we additionally tested using non-zero weight decay on the AdamW-updated parameter. The results (Fig. [linkB](https://anonymous.4open.science/r/NorMuon_ICML26_rebuttal-F9F8/weightdecay_NorMuon.png)) show that NorMuon’s advantage remains.
>
> For Muon-rescale, we sweep over different total step counts for Muon and identify the smallest step count whose validation loss matches that NorMuon target.
>
> > **Q3. How are the hyperparameters of Muon+Adam variant tuned?**
>
> We gave Muon+Adam a tuning budget comparable to NorMuon and selected its best-performing configuration. Additionally, please refer to our response to Reviewer yCE4 Q2 for a more detailed analysis and discussion of why Muon+Adam underperforms NorMuon.
>
> [1] Preconditioned Spectral Descent for Deep Learning
>
> [2] Training Deep Learning Models with Norm-Constrained LMOs
>
> [3] Muon is Scalable for LLM Training

---

> > ### Author Rebuttal · Reviewer_fGpc · 2026-04-03
> >
> > Thank you to the authors for the detailed and thoughtful rebuttal. I have also read the other reviews and your corresponding responses.
> >
> > Regarding the comparison between NorMuon and the Muon+Adam variant: while I appreciate the intuition and toy examples provided in your response to Reviewer yCE4, it remains somewhat counterintuitive to believe that NorMuon is universally better than Muon+Adam across all settings. I strongly encourage you to include a more in-depth ablation study (at least on a small model but with more tuning) and a deeper theoretical or empirical discussion comparing these two specific approaches in the final manuscript to fully substantiate this claim.

---

> > > ### Author Response · Authors · 2026-04-06
> > >
> > > We thank the reviewer for the thoughtful follow-up and constructive suggestion. We agree that claiming NorMuon is universally better than Muon+Adam would be too strong based on the current evidence. In the revision, we will therefore narrow our claim to the settings we have tested and expand the discussion of this comparison.
> > >
> > > To further address this point, we conducted an additional tuning study on the 124M setting. We report the sweep results below.
> > >
> > > **NorMuon**
> > >
> > > | Learning Rate | $\beta_1$ | $\beta_2$ | Val. Loss  |
> > > | :------------ | --------- | --------- | ---------- |
> > > | 0.00036       | 0.95      | 0.95      | **3.2996** |
> > > | 0.00018       | -         | -         | 3.3250     |
> > > | 0.00072       | -         | -         | 3.3100     |
> > > | 0.00144       | -         | -         | 3.7352     |
> > > | -             | 0.85      | -         | 3.3106     |
> > > | -             | 0.90      | -         | 3.3003     |
> > > | -             | -         | 0.90      | 3.3024     |
> > > | -             | -         | 0.99      | 3.3003     |
> > >
> > > **Muon+Adam**
> > >
> > > | Learning Rate | $\beta_1$ | $\beta_2$ | Val. Loss  |
> > > | ------------- | --------- | --------- | ---------- |
> > > | 0.00036       | 0.95      | 0.99      | **3.3011** |
> > > | 0.00018       | -         | -         | 3.3217     |
> > > | 0.00072       | -         | -         | 3.3047     |
> > > | 0.00144       | -         | -         | 3.4339     |
> > > | -             | 0.85      | -         | 3.3180     |
> > > | -             | 0.90      | -         | 3.3052     |
> > > | -             | -         | 0.90      | 3.3085     |
> > > | -             | -         | 0.95      | 3.3063     |
> > >
> > > Under a comparable sweep, NorMuon achieves a slightly better best validation loss than Muon+Adam at the optimal setting. At the same time, Muon+Adam appears to be more robust to learning-rate tuning, which may reflect the stabilizing effect of coordinate-wise scaling. Across different $\beta_1$ and $\beta_2$ we tested, however, NorMuon remains consistently competitive and typically better.
> > > Overall, we view these results as supporting a more precise claim: in the settings we tested, NorMuon matches or slightly outperforms Muon+Adam at its best tuned performance, which may be due to better preservation of the geometric benefits of Muon. We will revise the manuscript accordingly, include this expanded ablation, and add a more careful discussion of when each variant may be advantageous.

---

### Decision · Program_Chairs · 2026-04-30

**Decision:**

Accept (spotlight)

**Comment:**

This work proposes NorMuon, an optimizer leverages the strength of both AdamW and Muon, i.e. orthogonalization with neuron-level adaptive learning rates via row-normalization. The algorithm is easy to implement and to scale, as demonstrated on FSDP. The experiments are very promising in terms of loss values and downstream evaluations. The reviewers had some initial concerns about clarity, fair tuning, and missing baselines, which are adequately addressed during the rebuttal. Furthermore, multiple reviewers commented that this work lacks theoretical depth. In my opinion, this may be out of the scope and the empirical analysis is sufficiently significant. Nevertheless, the authors should include new material from the rebuttal in the final version.